## [Peer Review File · Nature Communications]

A potent SARS-CoV-2 neutralising nanobody shows therapeutic efficacy in the Syrian golden hamster model of COVID-19REVIEWER COMMENTS

Reviewer #1 (Remarks to the Author):

In this work, Huo et al. reported four nanobodies (C5, H3, C1, and F2) for the RBD of SARS-CoV-2 spike protein, and the Kds for these nanobodies were all in the picomolar range, structural studies also show that they bind to different epitopes. In addition, these nanobodies were engineered as homotrimers, which elicit higher affinity with spike protein. Especially, the C5-trimer showed potent therapeutic efficacy via the respiratory route in the Syrian hamster model of COVID-19 and separately effective prophylaxis. This is the continuous work after their NSMB paper (Neutralizing nanobodies bind SARS-CoV-2 spike RBD and block interaction with ACE2, *Nat Struct Mol Biol.* 2020 Sep;27(9):846-854). The data are well presented, and the work itself is rather comprehensive and provides detailed analysis from different aspects. However, may I say, the results are not technologically novel nor revealed neutralizing potency against newly emerging variants. I would like to suggest the author provide more information before the formal acceptance.

Major concerns:

1. The current figure 2 is misleading, the authors are suggested to redraw this figure, to better illustrate the structural details of the nanobody-RBD complex. The RBD molecule is suggested to be displayed as surface representation, and keep the same color across different panels, the binding epitopes of four nanobodies could be indicated on one RBD molecule.
2. Nanobodies reported in this work were engineered as homotrimers by joining the VHH domains with a glycine-serine flexible linker, and they elicit a much higher affinity with spike protein. However, the mechanism for this remains elusive, the cryoEM structure of C5 homotrimer with spike trimer would provide more details.
3. C5 in this work could bound to the inactive form of the spike trimer, and the authors infer that this inactive form was induced by C5, the results and evidence here are not sufficient to support this claim.
4. Nanobodies in this work elicit different affinity across different strains, in this work, Kent (B.1.1.7; N501Y) and South African (B1.351; N501Y, N417K, and E484K) were tested, have the authors test neutralizing activity against more emerging strains such as B.1.617 variant?

Minor concerns:

1. To date, many antibodies of different epitopes have been discovered, the authors are suggested to add a table to summarize the known epitopes in the supplementary section to acknowledge the contribution from other groups.
2. The epitopes recognized by H3 and H11-H4 have significant overlap, but there is an obvious contrast between the Kds for these two antibodies (Kd for H11-H4 is 44 nM). Would the authors add some structure comparison and prepare some variants to explain this?
3. H4-H11 on page 4, should it be H11-H4?

Reviewer #2 (Remarks to the Author):

The manuscript by Huo et al describes four nanobodies against SARS-CoV-2 that are able to neutralize wildtype virus. The authors characterize the biophysical characteristics via SPR/ITC and map the epitopes using X-ray crystallography. In addition, they characterize the nanobodies as trimers against a few variants of concern. While the C5 nanobody is not effective against B1.351, the authors proceed with it into in vivo models as it is the most potent against the B5 strain. Using C5 and its derivatives, they characterise the binding onto spike trimer using cryo-EM and deliver it as a prophylaxis or therapeutic both intranasally or via IP administration. Collectively this data complements the existing nanobody studies against SARS-CoV-2 that are already published and shows similar results to the recent study published in *Science Advances* by Nambulli and colleagues that inhaled nanobodies can prevent and treat severe COVID-19 in hamsters. Nevertheless, it is a good body of work that shows that nanobodies are good therapeutic options for development of respiratory delivery of drugs against COVID-19.

Major considerations:

1. The authors perform 2 rounds of phage panning with a 109 library, but only present data on 4 nanobodies which seems like a really low hit rate. It would be useful to have Supplementary Figure 1 as a summary of the screening process that includes a sequence alignment schematic of the four identified nanobodies (with CDRs and FRs labelled), number of clones identified and clonal analyses of the clones identified.
2. Could the naming of the variants throughout the manuscript be updated to WHO convention (alpha, beta, etc) and country-specific naming of the strains be removed.
3. Figure 2E is called out in the sentence "C1 does not occlude the ACE2 epitope but would sterically prevent ACE2 binding to RBD..." immediately following figure 2A and before Figure 2B. It also does not appear to be the correct reference figure for this statement. Could this be corrected, Figure 2A does appear to show what the authors describe here.
4. Figure 2C is used as a reference to compare the H3 nanobody to the H11-H4 nanobody in the sentence "The H3 interface involves CDR2 and CDR3 of the nanobody, similar to, but different from the H11-H4 nanobody." But figure 2C does not show the H11-H4 nanobody or allow a comparison to H3, could the figure be modified or a reference for H11-H4 be inserted here?
5. The authors state that "There is no contact with Y501 in the H3-Kent RBD so the explanation for the reduction in affinity must reflect subtle changes at the interface." Could this please be expanded upon to improve clarity. For example, are there any changes in other nanobody-RBD contacts or nanobody positioning in the N501Y structure that might be the associated with this "subtle change" or are any differences not visible in the structural data?
6. Figure 2D-E and Extended data Figure 3C-D are referenced in the sentence "The RBD epitopes recognized by C1 and F2 substantially overlap (Y369-A372, F374-T385 in common) but are not identical" however the overlap is perhaps more readily seen in figure 2A than the figures referenced, could 2A be added to the reference here.
7. The sentence "The comparison of the two nanobodies, shows that they are related by a pivot on a residue in CDR3" is somewhat ambiguous and should have a supporting figure to clarify the description.
8. Figure 2A is referenced in the sentence "Comparing the RBD structures in the various complexes shows that Y104 of F2 displaces the helix of RBD at Y369 by 3 Å (Fig. 2a)." But this figure does not show displacement of the RBD in F2 compared with the other complexes or the positioning of Y104 and Y369 in the structures. Could this displacement be added to the figure or depicted in an additional figure.
9. In Figure 3A, the positioning of the RBDs within the spike trimer is difficult to determine. Could these domains be highlighted/coloured differently and labelled to improve the clarity of the figure.
10. For Figure 3B the authors state "C5 (unlike H11-H4) cannot bind to the "1 up 2 down" active form due to steric clashes". However, it appears that Figure 3B is only a superposition of the nanobody onto a 1 up 2 down spike structure. Is there further experimental evidence to support this statement, and could it be provided here or the statement qualified to reflect that this is modelling?
11. In Figure 3B it is difficult to immediately identify which RBD the C5 nanobody is binding. Perhaps highlighting the relevant RBD or including chain colour of the bound RBD in the figure legend would make this easier for the reader.
12. In all in vivo data, the control hamsters received no treatment. Is there any possibility of using any available historical data that the authors may have for an isotype control?
13. In Figure 6b, for consistency, please label all statistical comparisons even if not significant.
14. In the Discussion (pg 15), the authors make a claim "first demonstration of a therapeutic treatment with a nanobody treatment" which is currently incorrect due to the publication of Nambulli et al in Science Advances. This statement should be removed.
15. Inclusion of results from Nambulli et al, Science Advances 2021 should be included where appropriate in the Introduction and the Discussion.
16. In general, throughout the results section, it was very hard to follow the results in relation to the figures from the lack of figure call outs in the text. This was particularly lacking in the sections of "Potent Neutralisation of SARS-COV-2 in vitro by trimeric nanobodies" and "C5-Fc fusion show therapeutic efficacy in vivo". Please add a figure reference to every result description.

Minor considerations:

1. In the introduction, the sentence “By far the most serious outbreak has been caused by SARS-CoV-2 which is responsible for the current global pandemic currently presently associated with >1.7 million deaths worldwide.” Should be updated to remove currently or presently and the number of deaths updated as this figure is considerably out of date.
2. In the Introduction, pg 3, please add reference 14 (Pymm and Adair et al, PNAS 2021) to the references for the sentences “...nanobodies against SARS-CoV-2 infection has been demonstrated in cell-based assays”. And also to “These include fusing to an Fc ...”
3. The first sentence in the structural analysis of RBD binding section lists the crystal structures and names one H3-RBD (Victoria)-C1, could this be renamed to H3-C1-RBD (Victoria) to make it easier for the reader to see it has both H3 and C1 nanobodies in complex with the RBD.
4. The first reference to Extended data Figure 2 should be to Extended data Figure 2A.
5. The sentence “Based on the structure, the H3 interaction would not be expected to be sensitive to the Kent mutation (N510Y) (Fig. 2c).” should be changed to (N501Y).
6. For the sentence “The smaller molecular size of the C5-trimer (40 kDa) compared to the C5-Fc (80 kDa plus 2N linked glycans) renders the nanobody suitable for respiratory administration directly to the airways.” Could a reference be provided to explain why an 80 kDa protein is not suitable for intranasal delivery?
7. The sentence “This decrease was significantly different in the group in the IP treated group” should be amended to remove duplication of “in the group”.
8. The statement “B1.1.7 lineage has become the dominant variant in the UK due to the fact that the N501Y change harbored by these viruses reduces the binding of some human antibodies.” Should be amended to reflect the current situation in the UK where the delta variant comprises over 99 % of new infections.
9. In the crystallography methods section please expand the abbreviation SPG and PEG when first used.
10. In the crystallography methods section 0.1 M Citrate pH 4 is given as part of a crystallisation condition, could the authors please specify the salt used here (e.g sodium citrate).
11. In the Cryo-EM methods section “equipped with a Bioquatum-K3 detector” should be amended to read Bioquantum.

Reviewer #3 (Remarks to the Author):

The manuscript by Huo and colleagues describes the production and characterization of four nanobodies engineered as homotrimers with high affinity to the SARS-CoV-2 RBD. Three nanobodies neutralize the Victoria strain and the highly transmissible B.1.1.7 Kent variant of concern, whereas only one of them neutralizes the B.1.351 South African variant of concern. The authors tested one of the three nanobodies in the Syrian hamster model showing that it is effective even though intranasal application.

Strength: Solid structural and in vitro characterization of nanobodies generated against the SARS-CoV-2 RBD. Interesting in vivo efficacy data using an intranasal application approach.

Weakness: Selection of nanobody for in vivo experiments and lack of infectious virus data as proof of efficacy.

Concerns:

- (1) Why did the authors not perform any combined nanobody treatment experiments (e.g. C1 & C5)? This would have likely increased efficacy in animals.
- (2) It would have strengthened the study if the authors would also have included efficacy studies with the two variants of concern to also demonstrate efficacy of their approach against these variants.
- (3) Important parameters for treatment efficacy such as subgenome RNA detection and infectious virus are either not shown (e.g., lung tissue) or shown as extended material (nasal washed;

oropharyngeal swabs). These objective measurements should get more weight by displaying them in the manuscript figures rather than showing subjective measure such as histopathology scorings. For example, infectious titers for lung and nasal tissues need to be shown in addition to ISH.

(4) The control groups in the in vivo experiments should have been treated with an unrelated nanobody rather than treated with PBS or being untreated.

(5) Figure 5: The histopathology slides should be enlarged. As presented, they don't really show a difference in ISH.

(6) Figure 6: A cartoon indicating the study design as done for Figure 5 would be helpful here.

RESPONSE TO REVIEWERS

Our responses are shown in italics. We thank the reviewers for their careful reading and constructive comments. We have addressed the critique by adding more data wherever possible and by improving the presentation of the reported results.

Reviewer #1 (Remarks to the Author):

In this work, Huo et al. reported four nanobodies (C5, H3, C1, and F2) for the RBD of SARS-CoV-2 spike protein, and the Kds for these nanobodies were all in the picomolar range, structural studies also show that they bind to different epitopes. In addition, these nanobodies were engineered as homotrimers, which elicit higher affinity with spike protein. Especially, the C5-trimer showed potent therapeutic efficacy via the respiratory route in the Syrian hamster model of COVID-19 and separately effective prophylaxis. This is the continuous work after their NSMB paper (Neutralizing nanobodies bind SARS-CoV-2 spike RBD and block interaction with ACE2, Nat Struct Mol Biol. 2020 Sep;27(9):846-854). The data are well presented, and the work itself is rather comprehensive and provides detailed analysis from different aspects. However, may I say, the results are not technologically novel nor revealed neutralizing potency against newly emerging variants. I would like to suggest the author provide more information before the formal acceptance.

Major concerns:

1. The current figure 2 is misleading, the authors are suggested to redraw this figure, to better illustrate the structural details of the nanobody-RBD complex. The RBD molecule is suggested to be displayed as surface representation, and keep the same color across different panels, the binding epitopes of four nanobodies could be indicated on one RBD molecule.

Figure 2 revised as suggested by the referee

2. Nanobodies reported in this work were engineered as homotrimers by joining the VHH domains with a glycine-serine flexible linker, and they elicit a much higher affinity with spike protein. However, the mechanism for this remains elusive, the cryoEM structure of C5 homotrimer with spike trimer would provide more details.

By linking three VHHs together we clearly gain an increase in binding to immobilised RBDs in our SPR assays due to the effect of avidity with each trimer binding to more than one RBD. The avidity gain from multimerisation is of course well known in biology.

We carried out two EM experiments, one in which the trimer of Spike was mixed 1:1.5 with trimer of C5 (that is one monomer to one monomer ratio). In a second experiment we mixed 1 spike trimer with three trimers of C5 (monomer ratio 1:3). The mixtures were applied to EM grids (essentially in the manner in the C5 monomer study in the manuscript). We screened these grids in a Glacios instrument. The grids did not show evidence of useful single particles, rather the images suggested ill-defined aggregation (please see Figures in Appendix 2 below). We have not pursued these further since we do believe they add to the central science message.

We assume that when neutralising the virus by blocking infection of cells in vitro, the C5 trimer binds to more than one RBD which could be from the same spike trimer or from adjacent spike proteins and most likely mixture of both. However, since we lack direct experimental evidence we prefer not to speculate on this in the manuscript.

3. C5 in this work could bound to the inactive form of the spike trimer, and the authors infer that this inactive form was induced by C5, the results and evidence here are not sufficient to support this claim.

The group of McLellan (Science) first observed the different forms of the Spike protein (all down, 1 up 2 down) in a landmark paper. They proposed that these forms existed in a dynamic equilibrium and this has been confirmed by subsequent studies. The referee is right to say that the "all down" form is thought to be incompetent for ACE2 binding in vivo. However, the equilibrium between states is thought to be rapid and it would be incorrect to assert that the native form of Spike is "inactive". A further study identified additional intermediate states (Gamblin, NSMB). Another important paper reported the presence of lipid that appeared to lock the Spike into an all down form (Berger Science). Interestingly this structure does not exhibit any equilibrium distribution on grids, thus the "locked all down" can be truly thought of as an inactive form. Yet another group showed that mutations in Spike could alter the equilibrium between forms. In the absence of the lipid or specific mutations or binding agent, the field has accepted that the Spike protein as purified will appear as multiple forms on the EM grids which are distinguishable. We ourselves noted multiple forms of the native protein as a confirmatory observation in our previous NSMB paper.

In the NSMB paper we observed that H11-H4 binding to Spike, results in a uniform conformation of 1 up 2 down. We reasoned that since the H11-H4 nanobody particularly favours this form (there is an additional structural interaction) then the equilibrium point for the complex is shifted to essentially all 1 up 2 down.

In the presence of C5, we uniformly see only the "all down" structure of Spike on our grids. Our maps are sufficient resolution to exclude the presence of lipid, in fact it is only the very careful protocol of Berger leads to the lipid form; we use the McLellan method. We are not dealing with the "locked all down" form. The ITC data show that the stoichiometry of C5 and Spike binding is 1:1 (monomer to monomer). We can therefore exclude the possibility that we are selecting a population of the Spike.

Simple modelling of the 1 up 2 down structure with C5 bound reveals that due to steric clashes between C5 on RBD with the neighbouring RBD in the Spike trimer, means that the 1 up 2 down Spike form of the trimer cannot bind to three C5 molecules (this figure is shown 4b).

We therefore conclude that the nanobody selectively binds to the "all down" form which is present along with other forms of the Spike in solution (established by McLellan and repeated by others including us). The very slow off rate of the nanobody means complex is no longer in equilibrium with the other forms of the Spike (that is the "all down" form is removed from the equilibrium). Le chateliers principle, means that a portion of 1 up 2 down form will convert to the "all down" form to re-establish the equilibrium. The "fresh" all down is then bound by C5 and the process repeats.

4. Nanobodies in this work elicit different affinity across different strains, in this work, Kent (B.1.1.7; N501Y) and South African (B.1.351; N501Y, N417K, and E484K) were tested, have the authors test neutralizing activity against more emerging strains such as B.1.617 variant?

The Delta variant only gained prominence after we had finalised the results for the paper and we have not yet had access to this virus for testing our nanobodies. However the two residues that are mutated in the RBD of this variant (L452R, T478K) are not involved in the interaction between any of the four nanobodies described and the RBD. Therefore we would expect that binding and neutralisation of the Delta variant would not be significantly affected. We have made this point in the discussion on page16. In a fast moving epidemic such as this with mutations arising, there is a balance between formal publication (which has timeframes and expectations of experimental thoroughness) and being maximally up to date. We believe we have chosen the right balance, this is an important piece of work, thoroughly executed which will advance the field.

Minor concerns:

1. To date, many antibodies of different epitopes have been discovered, the authors are suggested to add a table to summarize the known epitopes in the supplementary section to acknowledge the contribution from other groups.

We have updated the list of citations to include paper published since submission of our ms and also reference a very recent review that summarises the activity of many of the published anti-SARS-CoV-2 nanobodies.

2. The epitopes recognized by H3 and H11-H4 have significant overlap, but there is an obvious contrast between the Kds for these two antibodies (Kd for H11-H4 is 44 nM). Would the authors add some structure comparison and prepare some variants to explain this?

This is an insightful comment and we have expanded the text on page 6 to comment on the differences between H3 and H11-H4.

We now say

“The epitopes recognized by H3 and H11-H4 as we hypothesized do have a significant overlap (Fig. 3a). H3 however has 100 fold higher affinity than H11-H4. Since H3 and H11-H4 have quite different sequences and this results from many small changes in loops between the structure. This means that the identification of the atomic features that drive the difference in affinity from simple structural analysis is not straightforward. Comparison of the structures reveals several features that may contribute to the increased affinity The H3 RBD interface buries just under 10 % more surface area and satisfies 4 more hydrogen bonds than in H11-H4 RBD In addition, in H3 the key R52 E484 salt bridge makes additional hydrophobic interactions with W53 and F59 of H3 (Supplementary Fig. 3b), these contacts are absent in H11-H4. In a future study, we suggest these regions should be probed.”

A more systematic evaluation would require extensive site directed mutagenesis and testing. We suggest that such studies are beyond the scope of this paper and are not required to support the main scientific thrust of the paper (biological activity in hamsters 1 day post infection by topical administration).

3. H4-H11 on page 4, should it be H11-H4?

Corrected

Reviewer #2 (Remarks to the Author):

The manuscript by Huo et al describes four nanobodies against SARS-CoV-2 that are able to neutralize wildtype virus. The authors characterize the biophysical characteristics via SPR/ITC and map the epitopes using X-ray crystallography. In addition, they characterize the nanobodies as trimers against a few variants of concern. While the C5 nanobody is not effective against B1.351, the authors proceed with it into in vivo models as it is the most potent against the B5 strain. Using C5 and its derivatives, they characterise the binding onto spike trimer using cryo-EM and deliver it as a prophylaxis or therapeutic both intranasally or via IP administration. Collectively this data complements the existing nanobody studies against SARS-CoV-2 that are already published and shows similar results to the recent study published in Science Advances by Nambulli and colleagues that inhaled nanobodies can prevent and treat severe COVID-19 in hamsters. Nevertheless, it is a good body of work that shows that nanobodies are good therapeutic options for development of respiratory delivery of drugs against COVID-19.

Major considerations:

1. The authors perform 2 rounds of phage panning with a 109 library, but only present data on 4 nanobodies which seems like a really low hit rate. It would be useful to have Supplementary Figure 1 as a summary of the screening process that includes a sequence alignment schematic of the four identified nanobodies (with CDRs and FRs labelled), number of clones identified and clonal analyses of the clones identified.

Only 93 VHH clones were picked and analysed following two rounds of panning and screened by ELISA from which 15 unique VHH sequences were obtained. The four selected for study (C5, H3, F2 and C1) were the ones that showed the highest affinity binding to immobilised RBD as measured in an inhibition ELISA. These results and CDR sequences are provided in a new supplementary Figure 1 together with an alignment of the four selected nanobodies.

2. Could the naming of the variants throughout the manuscript be updated to WHO convention (alpha, beta, etc) and country-specific naming of the strains be removed.

We have now used the WHO convention and of course we rely on the PANGO lineage ids. In the scientific literature, the use of country where first detected is still commonly used shorthand. We would therefore prefer to retain (but de-emphasised form) these names. They can be removed if this is essential for publication.

3. Figure 2E is called out in the sentence "C1 does not occlude the ACE2 epitope but would sterically prevent ACE2 binding to RBD..." immediately following figure 2A and before Figure 2B. It also does not appear to be the correct reference figure for this statement. Could this be corrected, Figure 2A does appear to show what the authors describe here.

Corrected and our apologies.

4. Figure 2C is used as a reference to compare the H3 nanobody to the H11-H4 nanobody in the sentence "The H3 interface involves CDR2 and CDR3 of the nanobody, similar to, but different from the H11-H4 nanobody." But figure 2C does not show the H11-H4 nanobody or allow a comparison to H3, could the figure be modified or a reference for H11-H4 be inserted here?

The figure has been corrected, our apologies. We now include the following analysis on P6

"The epitopes recognized by H3 and H11-H4 as we hypothesized do have a significant overlap (Fig. 3a). H3 however has 100 fold higher affinity than H11-H4. Since H3 and H11-H4 have quite different sequences and this results from many small changes in loops between the structure. This means that the identification of the atomic features that drive the difference in affinity from simple structural analysis is not straightforward. Comparison of the structures reveals several features that may contribute to the increased affinity The H3 RBD interface buries just under 10 % more surface area and satisfies 4 more hydrogen bonds than in H11-H4 RBD In addition, in H3 the key R52 E484 salt bridge makes additional hydrophobic interactions with W53 and F59 of H3 (Supplementary Fig. 3b), these contacts are absent in H11-H4. In a future study, we suggest these regions should be probed. "

5. The authors state that "There is no contact with Y501 in the H3-Kent RBD so the explanation for the reduction in affinity must reflect subtle changes at the interface." Could this please be expanded upon to improve clarity. For example, are there any changes in other nanobody-RBD contacts or nanobody positioning in the N501Y structure that might be the associated with this "subtle change" or are any differences not visible in the structural data?

We do not observe any contacts with Y501 and H3. We cannot find an obvious structural explanation for the change in affinity. We are hesitant to speculate in the absence of a more systematic study the reason for the difference in affinity.

6. Figure 2D-E and Extended data Figure 3C-D are referenced in the sentence "The RBD epitopes recognized by C1 and F2 substantially overlap (Y369-A372, F374-T385 in common) but are not identical" however the overlap is perhaps more readily seen in figure 2A than the figures referenced, could 2A be added to the reference here.

Referenced now. We have also modified Figure 2, we now show the common epitope in a diagram of the RBD surface.

7. The sentence "The comparison of the two nanobodies, shows that they are related by a pivot on a residue in CDR3" is somewhat ambiguous and should have a supporting figure to clarify the description.

We have added a new Figure 3 (original in the extended data as Figure 2)) and modified the text on page 8 to clarify this point.

8. Figure 2A is referenced in the sentence "Comparing the RBD structures in the various complexes shows that Y104 of F2 displaces the helix of RBD at Y369 by 3 Å (Fig. 2a)." But this figure does not show displacement of the RBD in F2 compared with the other complexes or the positioning of Y104 and Y369 in the structures. Could this displacement be added to the figure or depicted in an additional figure.

A new panel has been added to Figure 2.

9. In Figure 3A, the positioning of the RBDs within the spike trimer is difficult to determine. Could these domains be highlighted/coloured differently and labelled to improve the clarity of the figure.

This has been done in the revised figure (now Figure 4a).

10. For Figure 3B the authors state "C5 (unlike H11-H4) cannot bind to the "1 up 2 down" active form due to steric clashes". However, it appears that Figure 3B is only a superposition of the nanobody onto a 1 up 2 down spike structure. Is there further experimental evidence to support this statement, and could it be provided here or the statement qualified to reflect that this is modelling?

We have qualified to make this clear that the basis of this statement that it cannot bind to the 1 up 2 down is based on modelling. In our reply to referee 1 we set out in more detail the arguments behind our analysis.

11. In Figure 3B it is difficult to immediately identify which RBD the C5 nanobody is binding. Perhaps highlighting the relevant RBD or including chain colour of the bound RBD in the figure legend would make this easier for the reader.

The revised Figure has these changes (now Figure 4)

12. In all in vivo data, the control hamsters received no treatment. Is there any possibility of using any available historical data that the authors may have for an isotype control?

We do not have our own isotype matched control data as the use of animals for the pilot experiment to test the C5-Fc was kept to the minimum (treatment plus placebo/vehicle control). We did this since Rogers et al., Science 369, 956–963 (2020) had previously shown in a study to evaluate human anti-SARS-CoV-2 RBD antibodies in the Hamster COVID-19 model that a human IgG1 isotype matched control had no therapeutic effect. Therefore ethically in our view the additional animal suffering required to repeat this control was not justified. We have added a comment to this effect on page 11.

13. In Figure 6b, for consistency, please label all statistical comparisons even if not significant.

Done

14. In the Discussion (pg 15), the authors make a claim "first demonstration of a therapeutic treatment with a nanobody treatment" which is currently incorrect due to the publication of Nambulli et al in Science Advances. This statement should be removed.

We have amended the text to include comment on the Nambulli et al paper. This was not sleight on our part, the paper was published after submission of our ms. We highlight that in our study treatment was 24 h after infection when first clinical signs of disease are manifested in the hamster model (Chan et al 2020 Clinical Infectious Diseases, 71,2428–2446,) which is a more demanding test

than the 6 h post infection reported by Nambulli et al. Overall both studies show the potential for nanobody agents in the treatment of COVID-19 by topical administration.

15. Inclusion of results from Nambulli et al, Science Advances 2021 should be included where appropriate in the Introduction and the Discussion.

The reference has been cited in the introduction and included in the discussion on page 15.

16. In general, throughout the results section, it was very hard to follow the results in relation to the figures from the lack of figure call outs in the text. This was particularly lacking in the sections of "Potent Neutralisation of SARS-COV-2 in vitro by trimeric nanobodies" and "C5-Fc fusion show therapeutic efficacy in vivo". Please add a figure reference to every result description.

We apologise and the referee is quite right, this has now been done

Minor considerations:

1. In the introduction, the sentence "By far the most serious outbreak has been caused by SARS-CoV-2 which is responsible for the current global pandemic currently presently associated with >1.7 million deaths worldwide." Should be updated to remove currently or presently and the number of deaths updated as this figure is considerably out of date.

Updated

2. In the Introduction, pg 3, please add reference 14 (Pymm and Adair et al, PNAS 2021) to the references for the sentences "...nanobodies against SARS-CoV-2 infection has been demonstrated in cell-based assays". And also to "These include fusing to an Fc ..."

Added

3. The first sentence in the structural analysis of RBD binding section lists the crystal structures and names one H3-RBD (Victoria)-C1, could this be renamed to H3-C1-RBD (Victoria) to make it easier for the reader to see it has both H3 and C1 nanobodies in complex with the RBD.

Done

4. The first reference to Extended data Figure 2 should be to Extended data Figure 2A.

Corrected

5. The sentence "Based on the structure, the H3 interaction would not be expected to be sensitive to the Kent mutation (N510Y) (Fig. 2c)." should be changed to (N501Y).

Corrected

6. For the sentence "The smaller molecular size of the C5-trimer (40 kDa) compared to the C5-Fc (80 kDa plus 2N linked glycans) renders the nanobody suitable for respiratory administration

directly to the airways." Could a reference be provided to explain why an 80 kDa protein is not suitable for intra-nasal delivery?

Reference added

7. The sentence "This decrease was significantly different in the group in the IP treated group" should be amended to remove duplication of "in the group".

Corrected

8. The statement "B1.1.7 lineage has become the dominant variant in the UK due to the fact that the N501Y change harbored by these viruses reduces the binding of some human antibodies." Should be amended to reflect the current situation in the UK where the delta variant comprises over 99 % of new infections.

This sentence has been removed and a comment on whether C5 would bind to the new Delta variant added.

9. In the crystallography methods section please expand the abbreviation SPG and PEG when first used.

Done

10. In the crystallography methods section 0.1 M Citrate pH 4 is given as part of a crystallisation condition, could the authors please specify the salt used here (e.g sodium citrate).

Done

11. In the Cryo-EM methods section "equipped with a Bioquatum-K3 detector" should be amended to read Bioquantum.

Done

Reviewer #3 (Remarks to the Author):

The manuscript by Huo and colleagues describes the production and characterization of four nanobodies engineered as homotrimers with high affinity to the SARS-CoV-2 RBD. Three nanobodies neutralize the Victoria strain and the highly transmissible B.1.1.7 Kent variant of concern, whereas only one of them neutralizes the B.1.351 South African variant of concern. The authors tested one of the three nanobodies in the Syrian hamster model showing that it is effective even though intranasal application.

Strength: Solid structural and in vitro characterization of nanobodies generated against the SARS-CoV-2 RBD. Interesting in vivo efficacy data using an intranasal application approach.

Weakness: Selection of nanobody for in vivo experiments and lack of infectious virus data as proof of efficacy.

Concerns:

(1) Why did the authors not perform any combined nanobody treatment experiments (e.g. C1 & C5)? This would have likely increased efficacy in animals.

We considered this and compared the neutralisation activity in vitro of C1-Fc and C5-Fc with an equimolar mixture of C1-Fc plus C5-Fc. The ND50 of the mixture was 0.06 nM compared to 0.7 nM and 0.029 nM for the individual fusion proteins. The potency of C5 is such that adding C1 merely dilutes the activity. On the basis of these results we concluded that treatment with a mixture would MOST LIKELY not have led to an increase in efficacy. Doing the experiment poses ethical issues for increased animal suffering.

(2) It would have strengthened the study if the authors would also have included efficacy studies with the two variants of concern to also demonstrate efficacy of their approach against these variants.

We have profiled the neutralisation activity in vitro of the nanobodies of the original SARS-CoV-2 virus (B Victoria stain) and on three variants and the in vivo efficacy of the most potent nanobody efficacy in animals. New variants will continue to arise faster than we can execute animal experiments, prepare thorough data for peer review and publish (rather than press release) our work. There are ethical concerns about testing every variants with every nanobody in living animals.

(3) Important parameters for treatment efficacy such as sub-genome RNA detection and infectious virus are either not shown (e.g., lung tissue) or shown as extended material (nasal washed; oropharyngeal swabs). These objective measurements should get more weight by displaying them in the manuscript figures rather than showing subjective measure such as histopathology scorings. For example, infectious titres for lung and nasal tissues need to be shown in addition to ISH.

Figure 5 (now Figure 6) has been revised and the viral titres, viral sg and RNA data previously presented in extended data exchanged for some of the histology and ISH results to give more prominence to the quantitative assessments of viral load. The histology and ISH results are now in Supplementary Figures 8 and 9.

Figure 6 (now Figure 7) has been revised as per the comments of referees 1 and 2 including an infographic of experimental design and the analysis of the data in panel (c) has been re-worked to include statistical comparisons between other pairs. With regard to referee #3, Intermediate nasal washed were not performed for this experiment. At necropsy (day 7 post-infection) none of the lungs contained infectious virus by plaque assay, likely due to the level being below the limit of detection for the plaque assay at the time point – this is consistent with our and other published data. A sentence to this effect has been added on page 14. However, as in figure 6b, there is qPCR data detailing virus load in lungs. This is the gold standard CDC assay that is being used for virus quantification in human and animal studies and as such we consider it the relevant assay in this context.

(4) The control groups in the in vivo experiments should have been treated with an unrelated nanobody rather than treated with PBS or being untreated.

Where there were no ethical constraints on animal use, this would be a control experiment. Based on our binding and neutralisation experiments and other published nanobody work, we were confident that the C5 nanobody bound very tightly and neutralised the virus. Unrelated nanobodies have no activity in binding or neutralisation experiments. The additional control of an unrelated nanobody would subject animals to suffering with no plausible expectation of effect. We therefore felt the vehicle (PBS) was the most appropriate sole control for this study.

(5) Figure 5: The histopathology slides should be enlarged. As presented, they don't really show a difference in ISH.

The histology and ISH results have been transferred to a revised Supplementary Figure 8 so that larger images can be presented.

(6) Figure 6: A cartoon indicating the study design as done for Figure 5 would be helpful here.

Added (now Figure. 7)

Appendices

1. Text of manuscript highlighted with changes/additions
2. EM images from C5_trimer: EM experiment

Copy of the text of the revised manuscript with changes highlighted

A potent SARS-CoV-2 neutralising nanobody shows therapeutic efficacy in the Syrian golden hamster model of COVID-19

Jiandong Huo^{1,2,4}, Halina Mikolajek⁵, Audrey Le Bas^{1,2,4}, Jordan J. Clark³, Parul Sharma³, Anja Kipar^{3,6}, Joshua Dormon^{1,4}, Chelsea Norman^{1,4}, Miriam Weckener¹, Daniel K. Clare⁵, Peter J. Harrison^{4,5}, Julia A. Tree⁷, Karen R. Buttigieg⁷, Francisco J. Salguero⁷, Robert Watson⁷, Daniel Knott⁷, Oliver Carnell⁷, Didier Ngabo⁷, Michael J. Elmore⁷, Susan Fotheringham⁷, Adam Harding⁸, Lucile Moynié¹, Philip N. Ward^{2,4}, Maud Dumoux¹, Yper Hall⁷, Julian A. Hiscox^{3,9,10}, Andrew Owen¹¹, William James⁸, Miles W. Carroll^{7,12}, James P. Stewart^{3,9,13}, *James H. Naismith^{1,2,4} and *Raymond J. Owens^{1,2,4}

¹Structural Biology, The Rosalind Franklin Institute, Harwell Science Campus, UK. ²Division of Structural Biology, The Wellcome Centre for Human Genetics, University of Oxford, UK. ³Department of Infection Biology & Microbiomes, Institute of Infection, Veterinary and Ecological Sciences, University of Liverpool, UK. ⁴Protein Production UK, The Rosalind Franklin Institute – Diamond Light Source, The Research Complex at Harwell, Science Campus, UK. ⁵Diamond Light Source Ltd, Harwell Science Campus, UK. ⁶Laboratory for Animal Model Pathology, Institute of Veterinary Pathology, Vetsuisse Faculty, University of Zurich, Switzerland. ⁷National Infection Service, Public Health England, Porton Down, Salisbury, UK. ⁸Sir William Dunn School of Pathology, University of Oxford, UK. ⁹Department of Preventive Veterinary Medicine, Northwest A&F University, Yangling, Shaanxi, China. ¹⁰Infectious Diseases Horizontal Technology Centre (ID HTC), A*STAR, Singapore. ¹¹Department of Pharmacology and Therapeutics, Centre of Excellence in Long-acting Therapeutics (CELT), University of Liverpool, UK. ¹²Nuffield Department of Medicine University of Oxford, UK. ¹³Department of Infectious Disease, University of Georgia, Georgia, USA.

*Corresponding authors

Abstract

SARS-CoV-2 remains a global threat to human health particularly as escape mutants emerge. There is an unmet need for effective treatments against COVID-19 for which neutralizing single domain antibodies (nanobodies) have significant potential. Their small size and stability mean that nanobodies are compatible with respiratory administration. We report four nanobodies (C5, H3, C1, F2) engineered as homotrimers with pmolar affinity for the receptor binding domain (RBD) of the SARS-CoV-2 spike protein. Crystal structures show C5 and H3 overlap the ACE2 epitope, whilst C1 and F2 bind to a different epitope. Cryo Electron Microscopy shows C5 binding results in an all down arrangement of the Spike protein. C1, H3 and C5 all neutralize the Victoria strain, and the highly transmissible Alpha (B.1.1.7 first identified in Kent, UK) strain and C1 also neutralizes the Beta (B.1.35, first identified in South Africa). Administration of C5-trimer via the respiratory route showed potent therapeutic efficacy in the Syrian hamster model of COVID-19 and separately effective prophylaxis. The molecule was similarly potent by intraperitoneal injection.

There are currently seven known coronaviruses that infect humans of which three (SARS-CoV-1, MERS, SARS-CoV-2) have emerged in the last 20 years and caused severe and even fatal respiratory diseases¹. By far the most serious outbreak has been caused by SARS-CoV-2 which is responsible for the current global pandemic currently presently associated with 3.94 million deaths worldwide. Although vaccines are now being administered against SARS-CoV-2, building up immunity in the global population will take time. The imperative to treat SARS-CoV-2 infection has led to the search for agents that neutralize the virus for use in passive immunotherapy. Early attention has focused on identifying neutralising monoclonal antibodies from patients who have recovered from COVID-19²⁻⁶; the therapeutic use of antibodies is widespread and draws on existing knowledge and resources. However, nanobodies or VHHs (Variable Heavy-chain domains of Heavy-chain antibodies) derived from the heavy chain-only subset of camelid immunoglobulins offer an alternative with multiple advantages over conventional antibodies. The small molecular size and stability of nanobodies allows them to be formulated for topical delivery directly to the airways of infected patients through aerosolization. This results in improved bioavailability, simpler therapeutic compliance and easier administration. Secondly, while conventional antibodies that comprise two disulphide-linked polypeptides, heavy and light chain, typically require mammalian cells for production, nanobodies can be manufactured using readily available microbial systems. The potency of nanobodies against SARS-CoV-2⁷ infection has been demonstrated in cell-based assays⁸⁻¹⁶ and most recently in animal studies^{17,18}. Several strategies for engineering VHH into a multivalent species are known. These include fusing to an Fc^{17,19-21} and simple N to C fusion of two or more nanobodies to the same epitope^{19,22}. Multivalent presentations increase the binding avidity to the molecular target and thus the biological potency of such agents²³. We have isolated four nanobodies that bind different epitopes on the receptor binding domain (RBD) of the SARS-CoV-2 spike (S) glycoprotein with high affinity and potently neutralize

the virus *in vitro* with picomolar potency. We have explored their binding to and neutralization of two newly emergent variants (B.1.1.7 and B.1.351), identifying a potent cross-reactive agent. We have shown that treatment either systemically (intraperitoneal route) or via the respiratory tract (intranasal route) with a single dose of the most potent nanobody prevented disease progression in the Syrian hamster model of COVID-19.

Results

Isolation and binding characterisation of nanobodies that block ACE2 binding to the Spike protein of SARS-CoV-2

Antibodies to the RBD of SARS-CoV-2 were raised in a llama by primary immunisation with a combination of purified RBD alone and RBD fused to human IgG1, followed by a single boost with purified S (spike) protein mixed with RBD. The S protein sequence was derived from the original Wuhan or Victoria (B) strain of SARS-CoV-2. A phage display VHH library was constructed from the cDNA of peripheral blood mononuclear cells, and RBD binders selected by two rounds of bio-panning. The phage clones with the highest affinity for RBD were identified by an inhibition ELISA and classified by sequencing of complementary determining region 3 (CDR3) (Supplementary Fig. 1). Four VHHs were selected for production and their RBD binding kinetics measured by surface plasmon resonance (SPR) (Fig. 1a-d). The calculated K_{DS} were all in the picomolar range (20 -615 pM) with the rank order of affinities H3>F2>C5>>C1 (Table 1).

Competition binding experiments were carried out by SPR to investigate whether the VHHs blocked the binding of RBD to ACE2 and the overlap with the epitope recognized by the human monoclonal antibody CR3022²⁴ as well as the nanobody H11-H4²⁵. The results showed that C1, H3 and C5 blocked ACE2 binding whereas F2 did not affect ACE2 binding (Fig. 1e). C1 and F2 but not C5 or H3 competed with CR3022 for binding to the RBD (Fig. 1f) whereas C5

and H3 but not C1 and F2 competed with H11-H4 binding (Fig. 1g). (CR3022 is known to recognize an epitope that does not overlap with ACE2²⁵⁻²⁷ or H4-H11²⁵). C5 and H3 would be expected to target a similar epitope to that of H11-H4, human monoclonal antibodies and other nanobodies that neutralise SARS-CoV-2 by competing directly with the interaction between the spike protein and the ACE2 receptor (cluster 2 antibodies²⁸). C1 and F2 belong to the group of antibodies (cluster 1 antibodies²⁸) including CR3022²⁶ and EY-6A²⁹ that bind to a region distinct from the ACE2 receptor binding interface. These two antibodies have been reported to destabilize the trimeric spike protein and by this mechanism prevent receptor engagement^{26,29} thereby neutralizing the virus.

ITC was used to analyse the binding of C5, F2 and C1 to RBD and spike proteins in solution. However, as the agents bind so tightly conventional ITC has large errors. Therefore a displacement assay was devised using the H11 nanobody previously identified²⁵ that weakly binds to RBD with a K_D of $1\ \mu\text{M}$ measured by ITC (Supplementary Fig. 2a). Combining the H11 titration with viral proteins (Supplementary Fig. 2a,b), C5 titration with viral proteins (Supplementary Fig. 2c,d) and C5 titration with viral proteins pre-incubated with H11 (displacement assay Supplementary Fig. 1e,f), we determined K_D for C5 to RBD as 210 ± 60 pM and to Spike as $350\ \text{pM} \pm 6\ \text{pM}$ (Supplementary Fig. 1g,h). The estimated K_D confirms sub-nanomolar binding of C5 to the Spike protein in solution and indicates 1:1 stoichiometry. No displacement agent was available for F2 and C1, and therefore the binding K_D for RBD of 320 ± 30 and 600 ± 40 pM respectively were estimated by direct binding but are subject to considerable uncertainty (Supplementary Fig. 1i,j). Both C1 and F2 when bound to Spike gave complex traces, suggesting that when engaging the Spike other conformational changes occur (Supplementary Fig. 1i,j).

The four nanobodies were also assessed for their binding to RBD from the Alpha (B.1.1.7; N501Y originally identified from the UK) and Beta (B.1.351; N501Y, N417K and E484K,

originally identified from South Africa). C5 and H3 bound strongly to the Alpha variant albeit with reduced affinity compared to the Victoria strain (Fig. 1 h,i) however, no binding was detected to the Beta strain. By contrast, C1 and F2 bound with a similar affinity to all three strains (Fig. 1). These results are consistent with the C5 and H3 epitopes overlapping with the mutated regions which are known to be adjacent to and part of the ACE2 binding region.

Structural analysis of RBD binding

To further define the epitopes recognized by the nanobodies, crystal structures of the C5-RBD (Victoria), H3-C1-RBD (Victoria) and F2-RBD (Victoria) co-complexes were determined to high resolution (Table 2, 1.5, 1.9 and 2.3 Å, respectively), however, the C1-RBD binary complex failed to give high quality crystals. Examination of the three structures confirmed the results of binding experiments that indeed H3 and C5 occlude the RBD binding site for ACE2 (Fig. 2a). C1 does not occlude the ACE2 epitope but would sterically prevent ACE2 binding to RBD, F2 would not be predicted to interfere with ACE2 binding (Fig. 2a). The C5 epitope has only a small overlap with the H3 epitope or with the H11-H4 epitope that we previously reported²⁵. The interface between C5 and RBD is extensive and involves all three CDR loops and the fixed sequence loop (FR2) at A75 of the nanobody (Fig. 2b and supplementary Fig. 3a).

The epitopes recognized by H3 and H11-H4 as we hypothesized do have a significant overlap (Fig. 3a). H3 however has 100 fold higher affinity than H11-H4. Since H3 and H11-H4 have quite different sequences and this results from many small changes in loops between the structure. This means that the identification of the atomic features that drive the difference in affinity from simple structural analysis is not straightforward. Comparison of the structures reveals several features that may contribute to the increased affinity The H3 RBD interface buries just under 10 % more surface area and satisfies 4 more hydrogen bonds than in H11-H4

RBD. In addition, in H3 the key R52 E484 salt bridge makes additional hydrophobic interactions with W53 and F59 of H3 (Supplementary Fig. 3b), these contacts are absent in H11-H4. In a future study, we suggest these regions should be probed.

The key binding interaction between C5 and H3 nanobodies and RBD is a combined salt bridge π -cation interaction involving an arginine from the nanobody (R31 in C5, R52 in H3) with E484 and F490 of RBD. This arrangement of the positively charged guanidine group, phenyl ring and glutamate was previously highlighted in the H11-H4 study²⁵. In C5, R31 is located in CDR1 and as result the side chain of R31 enters the salt bridge π -cation interaction from the opposite side to R52 but preserves the interaction (Fig. 3b). The E484K mutation found in the recently emergent South African and Brazilian strains will disrupt this interface in both C5 and H3 (as well as H11-H4). The formation of a salt bridge with E484 is a feature of many antibodies isolated from the B cells of COVID-19 convalescent and vaccinated individuals and escape mutants at this position are obviously a major concern for the efficacy of current vaccines^{30,31}.

In addition to R31, residues T28 to G30 from CDR1 of C5 are also in contact with residues Y453, L455, Q493 and S494 of RBD (Fig.2b and supplementary Fig. 3a). The aromatic ring of Y449 of the RBD makes extensive hydrophobic contacts with the main chain residues, T53 to G56 from CDR2 of C5. From C5 FR2 the main chain of S72, the side chains of N73 and N74 make hydrogen bonds with the side chains of Q498, N501 and the main chain of S494 respectively. The bidentate hydrogen bonding arrangement of N73 (from C5) with N501 explains why this interaction is sensitive to the N501Y mutation (Alpha variant). FR2 of C5 makes van der Waal interactions with Y449 and Y495 to G496 of the RBD. Finally, CDR3 residues V100, Y109 and F110 in C5 make van der Waals contacts with E484 to F486 of RBD (Fig.2b and supplementary Fig. 3a).

In H3, in addition to the R52 salt bridge, residues in CDR2 (R52 - F59) make either (or both) hydrogen bonds and van der Waals contacts with RBD (residues T470-I472, G482-E484 and F490) (Fig. 2c and supplementary Fig. 3a). From CDR3, I101 to Y106 make either (or both) hydrogen bonds and van der Waals contacts with RBD (Y449, L455, F456, E484, Y489, F490, L492-S494). Compared to the H11-H4 interaction, H3 has pivoted around V102 resulting in a shift of 2 Å at R52. It is this pivot that brings FR2 of H3 into contact with RBD (Fig.2b and supplementary Fig. 3a).

Based on the structure, the H3 interaction would not be expected to be sensitive to the mutation (N501Y) (Fig. 2c). The observation of the lower affinity of H3 for Alpha RBD is therefore surprising. In order to investigate this further the crystal structures of both H3 and C5 in complex with the Alpha RBD were determined. In neither the H3-RBD or H3-Alpha RBD complex is there any direct contact with residue 501. The crystal structures of these complexes do not reveal any differences in the nanobody RBD interface that result from the mutation. Molecular dynamics studies have identified that this mutation alters the dynamics of RBD and leads to an increase in affinity for ACE2³². It may be that altered dynamics are responsible for modifying the binding of H3. In the C5-Alpha RBD complex, N73 still makes a hydrogen bond interaction with Y501 but the arrangement is less geometrically ideal than with N501, consistent with the lower binding affinity observed (Fig. 3c).

The RBD epitopes recognized by C1 and F2 substantially overlap (Y369-A372, F374-T385 in common) but are not identical (Fig. 2a, f, and g and supplementary Fig. 3c,d). The C1 and F2 nanobodies are oriented differently, the relationship can be described as an approximate 40° rotation around residues 102 and 103 of CD3 (Fig. 2h). Interestingly this is very similar pivot point as we observed between H3 and H11-H4 (Fig. 3a). C1 buries more surface area and engages with several residues that are not contacted by F2 (G404-D405, V407, V503-G504, Y508). F2 meanwhile contacts L368, P412-Q414, D427-E429 that are not engaged by C1. C1

relies mainly on CDR3 (R100-W107, S109-S110, D112) with some contact with CDR2 (W50, S52, S54, D55, T57-T59) and one interaction with CDR1 (F31). The same regions are employed by F2 and once again CDR3 dominates (D99-Y105, R108, T110, E11, E113) followed by CDR2 (S52, W53, T56, P57, Y59) and one residue in CDR1 (T28). Comparing the RBD structures in the various complexes shows that Y104 of F2 displaces the helix of RBD at Y369 by 3 Å (Fig. 2i).

Residues T376- T385 of RBD also form part of the binding site of the VH domain of CR3022²⁶. Koenig et al¹¹ very recently reported two anti-RBD nanobodies (VHH_V and VHH_U) that bind in a similar location to C1 (and F2) and target this epitope (residues Y369-K378). On repeated passage of SARS-CoV-2 escape mutations were observed at these interface residues (Y369H, S371P, F377L and K378Q/N)¹¹, however actual variants incorporating these changes have yet to be identified³³.

In the context of the whole virus and from ultrastructural analysis of purified Spike by cryo-EM, RBD exists in an equilibrium of up and down conformations. Interaction between the spike protein and cell-surface ACE2 requires at least one RBD in the up or open conformation^{34,35}. The cryo-EM structure of the C5 bound to the spike protein (stabilised in the prefusion state³⁴) was determined by single particle cryo-EM (Table 3, Supplementary Fig. 4, and 5). C5 nanobodies were observed bound to the “3 down” (inactive)³⁶ form of the spike trimer (Fig. 4a). Simple modelling shows that C5 (unlike H11-H4) is unlikely to bind to the “1 up 2 down” active form due to steric clashes (Fig. 4b). We conclude that although C5 can only bind to the “all down” of the Spike, dynamic equilibrium between Spike conformers, results in the conversion to the “all down” complex. Other nanobody bound spike complexes have shown binding to either both up and down RBDs¹² or only up conformations¹¹. Incubation of C1 or F2 with the trimeric spike protein led to ill-defined aggregates on EM grids, indicating they destabilise the trimer, which would disrupt ACE2 engagement (Fig. S4). Similar findings were

reported for CR3022²⁶ and EY-6A²⁹ that recognize this epitope and are consistent with the complex ITC traces observed for binding of C1 and F2 to the spike protein in solution (Supplementary Fig. 2) This was attributed to the epitope being in the middle of the molecule and binding of a protein to this epitope is incompatible with the trimeric Spike structure.

Potent neutralisation of SARS-CoV2 *in vitro* by trimeric nanobodies

Linking more than one nanobody together to create bivalent and trivalent assemblies significantly increases antigen-binding due to avidity^{11,13,23,37-39}. Therefore, trivalent versions of the four nanobodies were constructed by joining the VHH domains with a glycine-serine flexible linker, (GS)₆. The nanobody homo-trimers (C5, C1 and H3) were produced by transient expression in expi293 cells and purified by metal chelate affinity chromatography and size exclusion. Although the F2 trimer was expressed it proved to be unstable on purification and was not pursued further. Binding of the trimeric nanobodies to the RBD was measured by SPR, and an approximate 10 to 100-fold enhancement in K_D was observed compared to the monomers (Table 1 and Supplementary Fig. 6). Notably, the H3 trimer was shown to have a sub-picomolar K_D for the RBD-Victoria with an off rate of approximately 6 hours. Binding of C5 trimer to RBD-Kent was shown to be only two-fold weaker than to RBD-Victoria, whilst binding of C5 monomer was ~25-fold weaker (Table 1, Fig. 1 and Supplementary Fig. 6).

Micro-neutralisation assays were carried out to test the effectiveness of the three nanobody trimers to block infection of Vero E6 cells by either Victoria, Alpha or Beta strains of the virus. All nanobodies potentially neutralized some if not all the strains (Fig. 5). Although H3 bound more tightly than C5 to the RBDs *in vitro*, it was less potent than C5 against both Victoria and Beta strains (Fig. 5b). Crucially, C5 was equipotent in neutralising these strains with IC₅₀s of 18 pM (Victoria - B) and 25 pM (Kent - B1.1.7) (Fig. 5b). As anticipated from the *in vitro* binding data, only C1 was active against the Beta (B1.351) strain (Fig. 5c).

The neutralization potency of the C5 trimer was confirmed in the Gold Standard Plaque Reduction Neutralisation Test (PRNT) against the Victoria strain which gave an ND50 of 3 pM (Supplementary Fig. 7)). This corresponds to one of the most potent neutralising nanobodies that has been identified to date^{10,13,39,40} and was therefore chosen to test for efficacy in an animal model of COVID-19.

C5-Fc fusion shows therapeutic efficacy *in vivo*

To probe neutralization *in vivo*, we tested C5 in the Syrian hamster model of COVID-19⁴¹⁻⁴³. As first demonstrated with SARS-CoV⁴⁴, Syrian hamsters are readily infectable, display both upper and lower respiratory tract viral replication, clinical signs and also pathological changes that are similar those seen in infected humans. Since an anti-MERS-CoV nanobody fused to immunoglobulin Fc fragment has previously shown to extend the half-life of the protein *in vivo* and ameliorate disease in a mouse challenge model⁴⁵ we first tested C5 as a huIgG1 Fc fusion protein. The RBD binding affinity (K_D 37 pM) and virus neutralisation potency (ND50 of 2 pM; 180 pg/ml) of C5-Fc was similar to the trivalent C5 protein, confirming the importance of multivalency for effective neutralisation (Table 1, Supplementary Fig. 6, 7). **Efficacy of a human IgG1 antibody has also been demonstrated in the Syrian hamster model with the isotype matched control showing no therapeutic effect⁶.**

The study comprised an experimental and a control group each of six animals. All animals in both groups were challenged intranasally (IN) with SARS-CoV-2 Victoria (5×10^4 pfu). The experimental group was treated 24 h later with a single dose of C5-Fc (4 mg /kg) administered intraperitoneally (IP) whilst the control group were left untreated (Fig. 6a). As a measure of disease progression, the animals were weighed each day over 7 days and nasal washes and oropharyngeal swabs were taken every other day (Fig. 6a). On day 7 the animals were culled and viral load in lung, trachea and duodenum measured by sub-genomic (sg)-RT-qPCR. Vital

organs were formalin-fixed for histopathology (H&E staining) and ISH RNAScope staining with SARS-CoV-2 S-gene probe to detect presence of virus RNA. SARS-CoV-2 infected animals exhibited progressive mean body weight loss (up to 17%) from day 1 to day 7 post challenge (pc) (Fig. 6b). In contrast, by day 7 post challenge (pc), animals in the nanobody treated group had lost significantly ($P < 0.005$, Mann Whitney) less weight (7%). High levels of nasal shedding of live virus (10^4 - 10^5 FFU/ml) were detected in 6/6 untreated animals (100%) on day 2 pc, whereas only 3/6 (50%) animals in the nanobody treated group shed virus (Fig. 6c). Some live viral shedding was seen in the throats of 3/6 control animals whereas no live virus was detected in the nanobody treated animals (0/6) on any day (Fig. 6c). Statistically significant lower levels of viral RNA were detected in throat swabs of treated compared to untreated controls on days 2, 4 and 7 pc (Fig 6e). However no difference in viral RNA was found in the nasal washes taken over the time course of the study or in homogenates of lung, trachea and duodenum following culling of the animals on day 7 (Fig. 6e and f). Measurements of sgRNA copies in either nasal washes, throat swabs and tissues showed no significant differences between the number of genomic copies of the virus between control and treated animals (Fig 6d and f).

Histopathology and RNAScope ISH techniques were used to compare the pathological changes and the presence of viral RNA in tissues from nanobody-treated and untreated control hamsters. A semiquantitative scoring system was combined with digital image analysis to calculate the area of lung with pneumonia and the quantity of virus. Viral RNA and lesions consistent with infection with SARS-CoV-2 were observed only in the nasal cavity (Supplementary Fig. 8) and lungs (Supplementary Fig. 9). No lesions were observed in any other organ studied. The lung lesions consisted of a bronchointerstitial pneumonia showing areas of parenchymal consolidation and were characterized by infiltration of macrophages and neutrophils, but also some lymphocytes and plasma cells (Supplementary Fig. 8c). The lesions in the nasal cavity

consisted in necrosis of the respiratory and olfactory mucosa and presence of inflammatory exudates and cell debris within the nasal cavity lumen. The area with pneumonia was significantly lower in the nanobody-treated hamsters together with a marked reduction of histopathology scores in the nasal cavity (Supplementary Fig. 9a). Statistically significant differences were also found for the presence of virus RNA in the lung or the nasal cavity (Supplementary Fig. 8b and 9b). Together, these results showed that a single therapeutic dose of C5-Fc administered IP reached the site of action in the lungs and nasal cavity and reduced viral load and associated pathological changes. Therefore, based on these promising results we undertook a larger study to evaluate the C5 trimer in the Syrian hamster model.

Trimeric C5 nanobody shows efficacy when administered via the respiratory route.

The smaller molecular size of the C5-trimer (40 kDa) compared to the C5-Fc (80 kDa plus 2N-linked glycans) renders the nanobody suitable for respiratory administration directly to the airways⁴⁶. Previously an anti-RSV nanobody trimer had been shown to be effective in reducing viral load in a disease model following intranasal delivery²³. Therefore, in the second animal study, the efficacy of the trimeric version of C5 was evaluated in the COVID-19 hamster model by administration using both IP and intranasal routes. The study consisted of five groups of six animals that were challenged with the SARS-CoV-2 strain Liverpool (1×10^4 pfu) on day 1 and weight changes followed over 7 days (Fig. 7a). To compare to the results obtained with the C5-Fc, the trimer was administered IP at 4 mg/kg; the same dose was delivered directly to the airways via intranasal installation (IN). A tenfold lower intranasal dose of 0.4 mg/kg of C5-trimer was also tested. As in the first study, animals in the untreated group showed a significant and progressive weight loss (20 % by day 7), whereas all animals treated therapeutically, 24 h after viral challenge, showed only a small weight loss and from day 2 had recovered to pre-challenged weights (Fig. 7b). The animals pre-treated 2 h before IN virus inoculation with 4

mg/kg C5 via the intranasal route showed no change in weight. The weight loss in all C5-treated groups was significantly different from the control group given PBS alone ($p < 0.01$; repeated measures two-way ANOVA). Analysis of viral load in the post-mortem lungs at day 7 by qPCR for Nucleoprotein (NP) RNA showed a decrease in the median value in treated compared to the untreated control animals. (Figure 7c). This decrease was significantly different in the IP treated group. While there was a clear trend in the other groups, there were two outliers with higher RNA load in each of the groups treated via the intranasal route. No live virus was detected by plaque assay in day 7 samples of lung homogenates consistent with what was observed in the first animal study (Fig 6c).

The histological and immunohistological examination showed multifocal extensive consolidation of the lung parenchyma in the untreated group, with multifocal patches of cells that expressed viral antigen (mainly type I and II pneumocytes, some cells morphologically consistent with macrophages) (Fig. 7d). The consolidated areas contained aggregates of macrophages and some neutrophils and were otherwise comprised of activated type II pneumocytes with occasional syncytial cell formation, and hyperplastic bronchiolar epithelial cells (Supplementary Fig. 10). In all treated groups, the extent of parenchymal consolidation was substantially reduced as quantified by automated morphometric analysis which resulted in a statistically-significantly larger area of ventilated lung parenchyma (Fig. 7d). The lungs of treated animals showed very limited viral antigen expression and only in occasional individual macrophages within small infiltrates or in pneumocytes in individual alveoli (Fig. 7e).

More detailed assessment of the consolidated areas in untreated animals confirmed that at day 7 post SARS-CoV-2 infection, the pathological processes in the lungs are dominated by regenerative attempts, as shown by type II pneumocyte and bronchiolar epithelial hyperplasia, in combination with macrophage dominated inflammatory infiltration (Supplementary Fig. 10). Animals that had received either C5-trimer (4 mg/kg) 2 h pre-infection or the lower dose

(0.4 mg/kg) at 4 h post infection, resulted in substantially less regenerative processes; the observed small, consolidated areas were dominated by infiltrating macrophages (Supplementary Fig. 10). These findings at the late, i.e., regenerative stage of SARS-CoV-2 infection in hamsters⁴² indirectly confirm that the C5-trimer treatment significantly reduced pulmonary infection and induced a strong macrophage response, likely leading to phagocytosis and thereby sequestration of the virus. Double immunofluorescence for viral N protein and the macrophage marker Iba1 undertaken on the lungs of hamsters that had been pre-treated with C5-trimer 2h prior to virus inoculation confirmed that numerous macrophages in the focal lesions contained viral antigen (Supplementary Fig. 11).

Collectively the animal studies described herein have established that a multivalent nanobody (Fc fusion or trimer) targeted to the RBD of SARS-CoV-2 spike protein delivered either systemically or via the respiratory route has a therapeutic benefit in the hamster disease model of COVID-19. In particular, efficacy was observed with a single IN dose of 0.4 mg/kg (equating to approximately 40 ug/ animal) of the C5-trimer demonstrating the high potency of this biological agent. A further dose ranging study will be required to establish the minimum amount of the nanobody required to be therapeutically effective in the hamster disease model.

Discussion

The RBD of SARS-CoV-2 is the immuno-dominant region of the virus spike protein and the target for neutralizing antibodies generated either by vaccination or infection. Following immunisation of a llama with a combination of the RBD and stabilised spike trimer³⁴ based on the Victoria strain sequence, we obtained nanobodies designated C5, F2, H3 and C1 that bound one of two orthogonal sites on the RBD. The site recognized by C5 and H3 overlapped with the ACE2 binding site on the top surface of the domain, whilst the second recognized by C1 and F2 corresponded to a location on the side of the RBD originally identified by the SARS-CoV antibody CR3022^{24,26,47} and nanobody VHH72⁴⁸. Consistent with other recent

reports^{10,17,39} nanobodies that bound to both sites showed very potent neutralization activity when configured as multivalent trimers, with the C5 trimer demonstrating complete inhibition of infection of Vero cells at < 100 pM in a PRNT assay. This activity was translated into a marked disease-modifying effect in the Syrian golden hamster model of COVID-19 with treated animals showing minimal weight loss and very limited pulmonary infection and associated changes following a single dose of C5 trimer 24 h post virus challenge. Most importantly, administration of the nanobody agent either directly by nasal administration or systemically (IP) was effective at 4.0 mg/kg. Nasal administration appeared to promote faster recovery than IP perhaps reflecting increased levels of the C5 trimer reaching the sites of infection in the lungs. Recently, mice challenged intranasally with SARS-CoV-2, and then treated prophylactically IP with a nanobody Fc fusion has also been shown to reduce viral load in the lungs¹⁷. More recently, Nebulli et al¹⁸, showed that nasal administration of a nanobody 6 h after viral challenge also reduced viral load and weight change in the Syrian hamster model. Our data are consistent with these results but our treatment with the C5 trimer 24 h after viral challenge when the clinical manifestations of disease first become apparent is a more demanding test of nanobody efficacy and arguably a more realistic model of therapeutic treatment.

The independent emergence of SARS-CoV-2 variants which appear to be more transmissible is now a major concern. Although in this study, animals were challenged with the Victoria and Liverpool (lineage B) strains, the *in vitro* neutralisation data strongly indicates the C5 trimer will be equally effective against the lineage B.1.1.7 or Alpha variant in this COVID-19 disease model. Although, the Alpha variant dominated infections in the UK in early 2021, the new the new Delta virus (B.1.671.2) that first originated in India has become the most recent variant of concern. The epitope recognised by C5 does not include the two residues that are mutated in the RBD of the Delta virus, L452R and T478K. However, F54 in Framework 3 of C5 does

make a Van der Waal interaction with L452 that may be disrupted by mutation to R452 (Supplementary Fig. 3). The B.1.351 (Beta variant) and P.1 (Gamma variant) lineages are characterized by three mutations (K417N, E484K and N501Y) in the RBD, which, although less prevalent, are a serious concern as they are associated with immune evasion³⁰. Structural analysis of the C5-RBD and H3-RBD complexes showed the central importance of E484 in RBD to the interaction and unsurprisingly these nanobodies failed to neutralize the Gamma virus. The C1 nanobody is significantly less potent than C5 against the Victoria strain, NT50 of C1 trimer is 4.9 nM compared to 18 pM and binds to a different epitope. However, C1 was equally effective against all three strains of the virus tested for neutralization *in vitro*, thus it has the potential to be a broadly neutralizing agent.

The relative size and stability of nanobody based bio-therapeutics has fueled interest in their use as inhaled drugs for the treatment of respiratory diseases⁴⁹, including for COVID-19⁵⁰. Furthermore, since some of their formulations, for example the trimeric molecule discussed here, do not require mammalian cell culture, they are relatively inexpensive to produce. In laboratory tests, anti-SARS-CoV-2 nanobody trimers, similar to the ones we report here, have already been shown to be stable under aerosolisation^{10,13}. Indeed, the trimeric anti-RSV nanobody (ALX-0171)²³, was successfully administered using a nebulizer in a Phase 1 safety study. This provides a useful precedent for developing locally administered products to treat respiratory viral illnesses. Local administration of nanobody therapy may not only treat disease but by reducing viral load, may rapidly and substantially lower infectivity.

In summary, we have identified a set of potent neutralizing SARS-CoV-2 nanobodies from an immunised llama library and mapped these onto the receptor binding domain of the spike protein. The two epitopes correspond to those targeted by human antibodies recovered from convalescent patients pointing to their cross species immunodominance. We show that SARS-CoV-2 infection in a hamster model can be treated with a single dose of the most potent

trimeric nanobody delivered either systemically or intranasally. Combinations of nanobodies that target different epitopes may improve resilience in combating new variants of the virus.

Methods

Immunisation and construction of VHH library

The SARS-CoV-2 receptor-binding domain (amino acids 330-532), SARS-CoV-2 receptor-binding domain fused to hIgG1 Fc (RBD-Fc) and trimeric spike protein (amino acids 1-1208) were produced as described by Huo et al 2020²⁵. Antibodies were raised in a llama by intramuscular immunization with 200 µg of recombinant RBD and 200 µg of RBD-Fc on day 0, and then 200 µg RBD and 200 µg S protein on day 28. The adjuvant used was Gerbu LQ#3000. Blood (150 ml) was collected on day 38. Immunizations and handling of the llama were performed under the authority of the project license PA1FB163A. Peripheral blood mononuclear cells were prepared using Ficoll-Paque PLUS according to the manufacturer's protocol; total RNA was extracted using TRIzol™; reverse transcription and PCR was carried out with SuperScript IV Reverse Transcriptase using primer CALL_GSP. The pool of VHH encoding sequences were amplified by two rounds of PCR using CALL_001 and CALL_02 (round 1), VHH_For and VHH_Rev_IgG2 plus VHH_Rev_IgG3 (round 2). Following purification by agarose gel electrophoresis, the VHH cDNAs were cloned into the SfiI sites of the phagemid vector pADL-23c. In this vector, the VHH encoding sequence is preceded by a pelB leader sequence followed by a linker, His6 and cMyc tag (GPPGGQHHHHHGGAEQKLI SEEDLS). Electro-competent

E. coli TG1 cells were transformed with the recombinant pADL-23c vector resulting in a VHH library of about 4×10^9 independent transformants. The resulting TG1 library stock was then infected with M13K07 helper phage to obtain a library of VHH-presenting phages.

Isolation of VHHs

Phages displaying VHHs specific for the RBD of SARS-CoV-2 were enriched after two rounds of bio-panning on 50 nM and 2 nM of biotinylated RBD respectively, through capturing with Dynabeads™ M-280 (Thermo Fisher Scientific). Enrichment after each round of panning was determined by plating the cell culture with 10-fold serial dilutions. After the second round of panning, 93 individual phagemid clones were picked, VHH displaying phages were recovered by infection with M13K07 helper phage and tested for binding to RBD by a combination of competition and inhibition ELISAs. In these assays, RBD was immobilized on a 96-well plate and binding of phage clones was measured in the presence of excess soluble RBD (inhibition ELISA) or the RBD-binding H11-H4-Fc²⁵ (competition ELISA).

Phage binders were ranked according to the inhibition assay and then classified as either competitive with H11-H4 (i.e., sharing the same epitope) or non-competitive (i.e. binding to a different epitope on RBD). Clones were sequenced and grouped according to CDR3 sequence identity.

Construction of trivalent VHHs

To generate the trimeric VHHs, the C1, C5, H3 and F2 gene fragments were used as templates to amplify three fragments by PCR with the following pairs of primers: TriNb_Neo_F1 and TriNb_R1; TriNb_F2 and TriNb_R2; TriNb_F3 and TriNb_Neo_R1; the three fragments were then joined together with a PCR reaction using primers TriNb_Neo_F2 and TriNb_Neo_R2. The trimeric gene product was then inserted into the pOPINTTGneo vector by Infusion® cloning. pOPINTTG contains a mu-phosphatase leader sequence and C-terminal His6 tag⁵¹.

Construction of receptor binding domain variants

To generate the RBD-Kent, using the RBD-WT as template, the gene was firstly amplified as two fragments with pairs of primers (1) TTGneo_RBD_F and N501Y_R and (2) TTGneo_RBD_R and N501Y_F; the two fragments were then joined together with a PCR

reaction using primer TTGneo_RBD_F and TTGneo_RBD_R. The RBD-Kent gene product was then cloned into the pOPINTTGneo vector by Infusion® cloning.

To generate the RBD-SA, using the RBD-Kent as template, the gene pre-RBD-SA was firstly amplified as two fragments with pairs of primers of (1) TTGneo_RBD_F and E484K_R and (2) TTGneo_RBD_R and E484K_F; the two fragments were then joined together with a PCR reaction using primer TTGneo_RBD_F and TTGneo_RBD_R. The pre-RBD-SA gene product was then cloned into the pOPINTTGneo vector by Infusion® cloning. Next, using the pre-RBD-SA as template, the gene RBD-SA was firstly amplified as two fragments with pairs of primers of (1) TTGneo_RBD_F and K417V_R and (2) TTGneo_RBD_R and K417V_F; the two fragments were then joined together with a PCR reaction using primer TTGneo_RBD_F and TTGneo_RBD_R. The RBD-SA gene product was then cloned into the pOPINTTGneo vector by Infusion® cloning.

To generate the huIgG1 Fc-fusion versions of RBDs, the RBD genes from the pOPINTTGneo vector were amplified by a pair of primers TTGneo_RBD_F and RBD_Fc_R, followed by being cloned into the pOPINTTGneo-Fc vector by Infusion® cloning. The pOPINTTGneo-Fc contains a mu-phosphatase leader sequence, a huIgG1 Fc and C-terminal His6 tag⁵¹

Protein production

In general, the monovalent VHHs were cloned into the vector pOPINO⁵² containing an OmpA leader sequence and C-terminal His6 tag. The C5 and H3 VHH constructs used for the crystallization of C5-Kent RBD and H3-Kent RBD complexes, respectively, were generated through amplification with a pair of primers PelB_F and PelB_R, followed by being cloned into the phagemid vector pADL-23c by Infusion® cloning. pADL-23c contains a PelB leader sequence and C-terminal His6 tag. The plasmids were transformed into the WK6 *E. coli* strain and protein expression induced by 1mM IPTG grown overnight at 20 °C. Periplasmic extracts were prepared by osmotic shock and VHH proteins purified by immobilised metal affinity

chromatography (IMAC) using an automated protocol implemented on an ÄKTXpress followed by a Hiloal 16/60 Superdex 75 or a Superdex 75 10/300GL column, using phosphate-buffered saline (PBS) pH 7.4 buffer. The C5-Fc was produced by transient expression in expi293® cells and purified by a combination of HiTrap MabSelect SuRe™ (Cytiva) and gel filtration in PBS pH 7.4 buffer. The trimeric versions of the nanobodies were produced by transient expression in expi293® cells and purified by a combination of IMAC and gel filtration in PBS pH 7.4 buffer. For animal studies, an additional ion exchange chromatography step was introduced after the IMAC (GE, Capto S 1mL column) to lower endotoxin levels which were further reduced to <0.1 EU/ml by passing in the final purified product through two Proteus NoEndo™ clean-up columns (Generon, Slough, UK). Endotoxin levels were quantified using the Pierce™ LAL Chromogenic Endotoxin Quantitation Kit (Thermofisher Scientific). Protein was concentrated to 4mg /ml and flash frozen for storage at -80 °C. The biotinylated and non-biotinylated RBDs, ACE2-Fc and CR3022-Fc were produced as previously described²⁵.

Surface plasmon resonance & ITC

The surface plasmon resonance experiments were performed using a Biacore T200 (GE Healthcare). All assays were performed with a running buffer of PBS pH 7.4 supplemented with 0.005% vol/vol surfactant P20 (GE Healthcare) at 25 °C.

The competition assay was performed with a Sensor Chip Protein A (Cytiva). CR3022-Fc, ACE2-Fc or H11-H4-Fc was used as the ligand, ~1,000 RU of CR3022-Fc, ACE2-Fc or H11-H4-Fc was immobilized. The following samples were injected: (1) a mixture of 1 µM nanobody C1 / C5 / H3/ F2 and 0.1 µM RBD-WT; (2) a mixture of 1 µM C2Nb6 (an anti-Caspr2 nanobody) and 0.1 µM RBD-WT; (3) 1 µM nanobody C1 / C5 / H3 / F2; (4) 1 µM C2Nb6; (5) 0.1 µM RBD-WT. All curves were plotted using GraphPad Prism 8.

To determine the binding kinetics between the SARS-CoV-2 RBD and nanobody C1 / C5 / H3 / F2, a Biotin CAPture Kit (Cytiva) was used. Biotinylated RBDs were immobilized onto the

sample flow cell of the sensor chip. The reference flow cell was left blank. Nanobody was injected over the two flow cells at a range of five concentrations prepared by serial two-fold dilutions, at a flow rate of $30 \mu\text{l min}^{-1}$ using a single-cycle kinetics program. Running buffer was also injected using the same program for background subtraction. All data were fitted to a 1:1 binding model using Biacore T200 Evaluation Software 3.1.

To determine the binding kinetics between the SARS-CoV-2 RBD-WT and C5-Fc, a Biotin CAPture Kit (Cytiva) was used. Biotinylated RBD was immobilized onto the sample flow cell of the sensor chip. The reference flow cell was left blank. C5-Fc was injected over the two flow cells at a single concentration of 10 nM, at a flow rate of $30 \mu\text{l min}^{-1}$. Running buffer was also injected using the same program for background subtraction. All data were fitted to a 1:1 binding model using Biacore T200 Evaluation Software 3.1.

To determine the binding kinetics between the SARS-CoV-2 RBD and the trimeric nanobodies C1/C5/H3, a Sensor Chip Protein A (Cytiva) was used. The huIgG1 Fc-fusion versions of RBDs were immobilized onto the sample flow cell of the sensor chip. The reference flow cell was left blank. Trimeric nanobody was injected over the two flow cells at a single concentration of 25 nM for C1 trimer, 10 nM for C5 trimer and 10 nM (RBD-Kent interaction) or 2.5 nM (RBD-WT interaction) for H3 trimer, at a flow rate of $30 \mu\text{l min}^{-1}$. Running buffer was also injected using the same program for background subtraction. All data were fitted to a 1:1 binding model using Biacore T200 Evaluation Software 3.1.

Isothermal titration calorimetry (ITC) measurements were carried out using an iTC200 and PEAQ-ITC MicroCalorimeter (GE Healthcare) at 25°C . RBD and all nanobodies were dialyzed into PBS and titrations into RBD were performed using 150 to 25 μM of nanobody and 14-2 μM RBD with the exception of Nb-H11 (470 μM) and RBD (47 μM). For spike protein, 80-60 μM nanobody were titrated into 8-6 μM spike (monomer concentration). Each experiment consisted of an initial injection of 0.4 μl followed by 16-19 injections of 2-2.4 μl nanobody into

the cell containing RBD or spike, while stirring at 750 rpm. For the displacement assays, approximately 200 μM of C5 nanobody was titrated into a mixture of 20 μM RBD and 100 μM H11 and 66 μM C5 nanobody was titrated into a mixture of 6 μM spike and 186 μM H11. Data acquisition and analysis were performed using the Origin scientific graphing and analysis software package (OriginLab) or AFFINImeter for global fitting of the displacement assay. For the fitting of C5 and H11 into spike, the monomeric concentration of spike and a single binding mode have been used. Data analysis was performed by generating a binding isotherm and best fit using the following parameters: N (number of sites), ΔH (kJmol^{-1}), ΔS ($\text{JK}^{-1}\text{mol}^{-1}$), and K (binding constant in molar^{-1}). Following data analysis, K was converted to the dissociation constant (Kd).

Determination of the structure of VHH- RBD complexes by X-ray crystallography

Purified VHHS were mixed with de-glycosylated RBD at a molar ratio of 1.2:1, and the complex purified by size exclusion chromatography as described⁸. The optimal conditions for crystallization of each complex were F2-RBD 0.1M Succinic Acid, Sodium Dihydrogen Phosphate and Glycine (SPG), pH 8, 25 % Polyethylene glycol (PEG) 1500, H3-C1-RBD and H3-C1-Alpha RBD 1.0 M Lithium chloride, 0.1 M Citric acid pH 4, 20 % PEG 6000 and C5-RBD 0.2 M Sodium Acetate, 0.1 M Sodium Cacodylate pH 6.5, 30 % w/v PEG 8000 and the C5-Alpha RBD 0.2 M Ammonium fluoride and 20 % PEG 3350. The protein concentrations for all complexes were 18 mg/ml except for F2-RBD, where 34 mg/ml was used. Crystals were grown at 20 °C by sitting drop vapour diffusion method by mixing 0.1 μl of protein complex (C5-RBD) with 0.1 μl of reservoir; mixing 0.2 μl of protein complex (F2-RBD; H3-C1-RBD) with 0.1 μl of reservoir or 0.1 μl of protein complex (C5-Alpha RBD; H3-C1-Alpha RBD) and 0.2 μl of reservoir as stated above. Crystals were cryoprotected with 30 % glycerol, cryocooled in liquid nitrogen, diffraction data collected and processed at the beamlines I03, I04 and I24 of

Diamond Light Source, UK. The structures were solved by molecular replacement using the H11-H4 RBD structure as the search model.

Cryo-EM structures

Preparation of cryo-EM grids, data collection and processing were carried out as previously described⁸. Briefly, purified spike protein in 10 mM Hepes, pH 8, 150 mM NaCl, at 1 mg/ml was incubated with nanobody C5, purified in PBS, at a molar ratio of 1:1.2 (Spike monomer:nanobody) at 16°C overnight. SPT Labtech prototype 300 mesh 1.2/2.0 nanowire grids were glow-discharged on low for 4 min (Plasma Cleaner PDC-002-CE, Harrick Plasma) and used in a Chameleon EP system (SPT Labtech) at 80% relative humidity, ambient temperature. Frozen grids were screened, and data collected using Titan Krios G2 (Thermo Fisher Scientific) equipped with a Bioquantum-K3 detector (Gatan, UK) operated at 300 kV. Data collection statistics are given in Supplementary Table 3. The RELION_IT.py processing pipeline as implemented in eBIC was used for automatic data processing up to 2D classification. The data were first processed as C1 but as the complex showed C3 symmetry, this was later changed to C3. The best 3D class was selected for further refinement, CTF refinement, and particle polishing within Relion. An initial model based on PDB ID 6VXX was created and the RBD-C5 crystal structure placed into density. The final model with correlation coefficient 0.76 was generated by multiple cycles of manual intervention in coot⁵³ followed by jelly body refinement using RefMac5 via CCP-EM GUI^{53,54}. Model validation was carried out in PHENIX⁵⁴⁻⁵⁶. Data processing and refinement statistics are given in Table 3.

Micro-neutralisation assay

VHH trimers were serially diluted into Dulbecco's Modified Eagles Medium (DMEM) containing 1 % (w/v) foetal bovine serum (FBS) in a 96-well plate. SARS-CoV-2 strains (B VIC01, B1.17 and B1.351) passage 4 (Vero 76) [9×10^4 pfu/ml] diluted 1:5 in DMEM-FBS

were added to each well with media only as negative controls. After incubation for 30 min at 37 °C, Vero cells (100 µl) were added to each well and the plates incubated for 2 h at 37 °C. Carboxymethyl cellulose (100 µl of 1.5 % v/v) was then added to each well and the plates incubated for a further 18-20 h at 37 °C. Cells were fixed with paraformaldehyde (100 µl /well 4 % v/v) for 30 min at room temperature and then stained for SARS-CoV-2 nucleoprotein using a human monoclonal antibody (EY2A). Bound antibody was detected by incubation with a goat anti-human IgG HRP conjugate and following substrate addition imaged using an ELISPOT reader. The neutralization titer was defined as the titer of VHH trimer that reduced the Foci forming unit (FFU) by 50% compared to the control wells.

PRNT assay

Plaque reduction neutralization tests (PRNT) were carried out at Public Health England using SARS-CoV-2 (hCoV-19/Australia/VIC01/2020) (GISAID accession number EPI_ISL_406844) generously provided by The Doherty Institute, Melbourne, Australia at P1 and passaged twice in Vero/hSLAM cells [ECACC 04091501]. Virus was diluted to a concentration of 933 p.f.u. ml⁻¹ (70 p.f.u./75 µl) and mixed 50:50 in minimal essential medium (MEM; Life Technologies) containing 1 % FBS (Life Technologies) and 25 mM HEPES buffer (Sigma) with doubling antibody dilutions in a 96-well V-bottomed plate. The plate was incubated at 37 °C in a humidified box for 1 h to allow neutralization to take place. Afterwards, the virus-antibody mixture was transferred into the wells of a twice Dulbecco's PBS-washed 24-well plate containing confluent monolayers of Vero E6 cells (ECACC 85020206, PHE) that had been cultured in MEM containing 10 % (v/v) FBS. Virus was allowed to adsorb onto cells at 37 °C for a further hour in a humidified box, then the cells were overlaid with MEM containing 1.5 % carboxymethyl cellulose (Sigma), 4 % (v/v) FBS and 25 mM HEPES buffer. After five days incubation at 37 °C in a humidified box, the plates were fixed overnight with 20 % formalin/PBS (v/v), washed with tap water and then stained with 0.2 % crystal violet

solution (Sigma) and plaques were counted. A mid-point probit analysis (written in R programming language for statistical computing and graphics) was used to determine the dilution of antibody required to reduce SARS-CoV-2 viral plaques by 50 % (ND50) compared with the virus-only control (n = 5). The script used in R was based on a previously reported source script⁴⁴. Antibody dilutions were run in duplicate and an internal positive control for the PRNT assay was also run in duplicate using a sample of heat-inactivated (56 °C for 30 min) human MERS convalescent serum (pH 7.4, 137 mM NaCl, 1 mM CaCl₂) and 1 mg ml⁻¹ trypsin (Sigma-Aldrich) to neutralize SARS-CoV-2 (National Institute for Biological Standards and Control, UK).

Evaluation of C5-Fc efficacy in the Syrian hamster model (Public Health England)

Golden Syrian hamsters (*Mesocricetus auratus*) (males and females) aged between 7-9 weeks old, weighing 110-140g, were obtained from Envigo, London, UK. Hamsters were assigned randomly and housed in individual cages with access to food and water ad libitum. All experimental work was conducted under the authority of a UK Home Office approved project license that had been subject to local ethical review at PHE Porton Down by the Animal Welfare and Ethical Review Body (AWERB) as required by the 'Home Office Animals (Scientific Procedures) Act 1986'.

Twelve hamsters were briefly anesthetized with 5 % isoflurane (Zoetis, Leatherhead, UK) and 4L/m O₂ and inoculated by the intranasal route with 5 x 10⁴ p.f.u./animal of SARS-CoV-2 (hCoV-19/Australia/VIC01/2020) delivered in 100 µl per nostril (200 µl in total). At day 1 post-challenge (pc) 6 hamsters were treated with 4 mg/kg of C5 Nanobody via the intraperitoneal route. Control hamsters (n=6) received no treatment. Temperature (taken using a microchip reader and implanted temperature/ID chip) and clinical signs were monitored twice daily, weight once daily. Clinical signs were scored as follows; healthy = 0, behavioral changes = 1, ruffled fur = 2, wet tail = 2, dehydrated = 2, eyes shut = 3, arched back = 3, wasp waisted

= 3, labored breathing = 5. Clinical samples of nasal washes in Dulbecco's PBS (DPBS, Gibco) (200 µl) as well as oropharyngeal (throat) swabs (MWE, Corsham, UK) were obtained prior to infection (day -2) and on days 2, 4, 6 and 7 pc; animals were briefly anesthetized for the collection of these samples. On day 7 all the hamsters were euthanized by an overdose of anesthetic (sodium pentobarbitone [Dolelethal, Vetquinol UK Ltd]) via the intraperitoneal route. At necropsy nasal washes and oropharyngeal swabs and tissue samples (lung, trachea and duodenum) were collected in PBS and stored frozen at -80°C for viral RNA measurement and viral culture. Tissue samples for histopathological examination were fixed in 10% buffered formalin at room temperature (see below).

A micro-plaque assay⁵⁷ was used to determine the amount of virus in tissue samples. The animal sample was serially diluted in assay diluent (MEM supplemented with L-glutamine (Life Technologies), non-essential amino acids (Life Technologies), 25mM HEPES (Sigma) and 1x antibiotic/antimycotic) and added to confluent monolayers of Vero E6 cells. The virus was adsorbed to the cells for 1 hr at 37°C. The inoculas were removed from the cell plates and a viscous overlay (1% carboxymethylcellulose, Sigma) was added. The plates were then incubated for 24 hr at 37°C. The cells were then fixed using 8 % formalin for >8 hrs and an immunostaining protocol was performed on the fixed cells (Bewley et al, 2021). Stained foci [foci forming units (FFU)] were counted using an ELISpot counter (Cellular Technology Limited, USA). The counted foci data was then plotted using Graph Pad version 9. A SARS-CoV-2 positive control at 1×10^5 PFU/ml was run alongside the animal samples, on each assay plate, with uninfected assay diluent as negative control.

RNA was isolated from nasal washes, oropharyngeal swabs and tissue samples (lung, trachea and duodenum). Weighed tissue samples were homogenized and inactivated in RLT (Qiagen) supplemented with 1% (v/v) beta-mercaptoethanol. Tissue homogenate was then centrifuged through a QIAshredder homogenizer (Qiagen) and supplemented with ethanol as per

manufacturer's instructions. Downstream extraction was then performed using the BioSprint™96 One-For-All vet kit (Indical Bioscience) and Kingfisher Flex platform as per manufacturer's instructions. Non-tissue samples were inactivated in AVL (Qiagen) and ethanol, with final extraction using the BioSprint™96 One-For-All vet kit (Indical Bioscience) and Kingfisher Flex platform as per manufacturer's instructions. Reverse transcription-quantitative polymerase chain reaction (RT-qPCR) was performed using TaqPath™ 1-Step RT-qPCR Master Mix, CG (Applied Biosystems™), 2019-nCoV CDC RUO Kit (Integrated DNA Technologies) and QuantStudio™ 7 Flex Real-Time PCR System. Sequences of the N1 primers and probe were: 2019-nCoV_N1-forward, 5' GACCCCAAATCAGCGAAAT 3'; 2019-nCoV_N1-reverse, 5' TCTGGTACTGCCAGTTGAATCTG 3'; 2019-nCoV_N1-probe, 5' FAM-ACCCCGCATTACGTTTGGTGGACC-BHQ1 3'. The cycling conditions were 25°C for 2 min, 50°C for 15 min, 95°C for 2 min, followed by 45 cycles of 95°C for 3 seconds, 55°C for 30 seconds. The quantification standard was in vitro transcribed RNA of the SARS-CoV-2 N ORF (accession number NC_045512.2) with quantification between 10 and 1x10⁶ copies/μl. Positive samples detected below the lower limit of quantification (LLOQ) of 10 copies/μl were assigned the value of 5 copies/μl, undetected samples were assigned the value of 2.3 copies/μl, equivalent to the assays LLOD. For nasal wash and oropharyngeal swab extracted samples this equates to an LLOQ of 1.29 x10⁴ copies/mL and LLOD of 2.96 x10³ copies/mL. Samples detected between LLOQ and LLOD were assigned 6.43 x10³ copies/mL. For tissue samples this equates to an LLOQ of 1.31x10⁴ copies/g and LLOD of 5.71 x10⁴ copies/g. Samples detected between LLOQ and LLOD were assigned 2.86 x10⁴ copies/g. Subgenomic RT-qPCR was performed on the QuantStudio™ 7 Flex Real-Time PCR System using TaqMan™ Fast Virus 1-Step Master Mix (Thermo Fisher Scientific) and oligonucleotides as specified by Wolfel et al⁵⁸., with forward primer, probe and reverse primer at a final concentration of 250 nM, 125 nM and 500 nM respectively. Sequences of the sgE

primers and probe were: 2019-nCoV_sgE-forward, 5' CGATCTCTTGTAGATCTGTTCTC 3'; 2019-nCoV_sgE-reverse, 5' ATATTGCAGCAGTACGCACACA 3'; 2019-nCoV_sgE-probe, 5' FAM- AACTAGCCATCCTTACTGCGCTTCG-BHQ1 3'. Cycling conditions were 50°C for 10 minutes, 95°C for 2 min, followed by 45 cycles of 95°C for 10 seconds and 60°C for 30 seconds. RT-qPCR amplicons were quantified against an in vitro transcribed RNA standard of the full-length SARS-CoV-2 E ORF (accession number NC_045512.2) preceded by the UTR leader sequence and putative E gene transcription regulatory sequence described by Wolfel et al. in 2020⁴⁹. Positive samples detected below the lower limit of quantification (LLOQ) were assigned the value of 5 copies/μl, whilst undetected samples were assigned the value of ≤0.9 copies/μl, equivalent to the lower limit of detection of the assay (LLOD). For nasal washes and oropharyngeal swabs extracted samples this equated to an LLOQ of 1.29x10⁴ copies/mL and LLOD of 1.16x10³ copies/mL. For tissue samples this equates to an LLOQ of 5.71x10⁴ copies/g and LLOD of 5.14x10³ copies/g.

The lung, nasal cavity including olfactory and respiratory mucosa, heart, liver, spleen, pancreas, trachea/larynx brain and small intestine (duodenum) were taken from each animal and were fixed in 10% neutral-buffered formalin, processed, embedded in paraffin wax and 4 μm thick sections cut and stained with haematoxylin and eosin (H&E). The tissue sections were digitally scanned and reviewed by a qualified veterinary pathologist blinded to treatment and group details and the slides were randomised prior to examination in order to prevent bias (blind evaluation). A scoring system was used to evaluate objectively the histopathological lesions observed in the tissue sections: 0=within normal limits; 1=minimal; 2=mild; 3=moderate and 4=marked/severe. Moreover, the area of the lung with pneumonia was calculated using digital image analysis (Nikon-NIS-Ar software package).

RNAscope (an in-situ hybridisation method used on formalin-fixed, paraffin-embedded tissues) was used to identify the SARS-CoV-2 virus in all tissues. Briefly, tissues were pre-

treated with hydrogen peroxide for 10 mins at room temperature (RT) target retrieval for 15 mins (98-101 °C) and protease plus for 30 mins (40 °C) (all Advanced Cell Diagnostics). A V-nCoV2019-S probe (Advanced Cell Diagnostics) targeting the S-protein gene was incubated on the tissues for 2 hours at 40°C. Amplification of the signal was carried out following the RNAscope protocol (RNAscope 2.5 HD Detection Reagent – Red) using the RNAscope 2.5 HD red kit (Advanced Cell Diagnostics). Appropriate controls were included in each ISH run. Digital image analysis was carried out with the Nikon NIS-Ar software package in order to calculate the total area of the tissue section positive for viral RNA. The images were scanned digitally using a Hamamatsu NanoZoomer S360 digital slide scanner and examined using Ndp.view2 v2.9.22 software. Nikon NIS-Ar software was used to perform digital image analysis in order to quantify the presence of viral RNA in lung sections. Graph and statistical analysis were performed with Graphpad Prism 9 and Minitab version 16.

Evaluation of C5 trimer therapeutic efficacy in the Syrian hamster model (University of Liverpool)

Animal work was approved by the local University of Liverpool Animal Welfare and Ethical Review Body and performed under UK Home Office Project Licence PP4715265. Male golden Syrian hamsters (8-10 weeks old) were purchased from Janvier Labs (France). Animals were maintained under SPF barrier conditions in individually ventilated cages. For virus infection the Liverpool strain was used, a PANGO lineage B strain of SARS-CoV-2 (hCoV-2/human/Liverpool/REMRQ0001/2020)⁵⁹. Animals were randomly assigned into multiple cohorts of 6 animals. For SARS-CoV-2 infection, hamsters were anaesthetised lightly with isoflurane and inoculated intra-nasally with 100 µl containing 10⁴ PFU SARS-CoV-2 in PBS. Hamsters were treated with 100 µl via either the intraperitoneal or intranasal route with C5 trimer contained in PBS. Animals were sacrificed at variable time-points after infection by an overdose of pentobarbitone. Tissues were removed immediately for downstream processing.

From all animals the left lung was fixed in 10% buffered formalin for 48 h and then stored in 70% ethanol until further processing. Two longitudinal sections were prepared and routinely paraffin wax embedded. Consecutive sections (3-5 μm) were prepared and stained with HE for histological examination or subjected to immunohistological staining. Immunohistology was performed to detect SARS-CoV-2 antigen, macrophages (Iba1+), type II pneumocytes (SP-C+) and epithelial cells (pan-cytokeratin+), using the horseradish peroxidase (HRP) method and the following primary antibodies: rabbit anti-SARS-CoV nucleocapsid protein (Rockland, 200-402-A50), rabbit anti-human Iba1/AIF1 (Wako, 019-19741), rabbit anti-human prosurfactant protein-C (SP-C; Abcam, ab40879), and mouse anti-human pan-cytokeratin (clone PCK-26; Novus Biologicals, NB120-6401). Briefly, after de-paraffination, sections underwent antigen retrieval in citrate buffer (pH 6.0; Agilent) (anti-SARS-CoV-2, -Iba1) or Tris-EDTA buffer (pH 9.0) (anti-SP-C, -pan-cytokeratin) for 20 min at 98 °C and for 20 min at 37 °C respectively, followed by incubation with the primary antibody overnight at 4 °C (anti-SARS-CoV, -SP-C) or 60 min at RT (anti-Iba1, -pan-cytokeratin). This was followed by blocking of endogenous peroxidase (peroxidase block, Agilent) for 10 min at room temperature (RT) and incubation with the secondary antibody, EnVision+/HRP, Rabbit and Mouse respectively (Agilent) for 30 min at RT, followed by EnVision FLEX DAB+ Chromogen in Substrate buffer (Agilent) for 10 min at RT, all in an autostainer (Dako). Sections were subsequently counterstained with haematoxylin. The anti-Iba1, -SP-C and -pan-cytokeratin antibodies were tested for their cross reactivity in hamster tissues, using the lung of an uninfected control hamster as positive control. For double immunofluorescence, sections underwent antigen retrieval in citrate buffer (pH 6.0) and were then incubated with the first primary antibody (rabbit anti-SARS-CoV), overnight at 4 °C, followed by blocking of the endogenous peroxidase (see above) and 1 h incubation with the red fluorescence labelled antibody (goat anti-rabbit 594; Invitrogen, A11012), incubation with the second primary antibody (goat anti-human Iba1; Abcam, ab 5076), overnight at 4 °C,

and 1 h incubation with the green fluorescence labelled antibody (donkey anti-goat 488; Invitrogen, A1105). The final incubation was with DAPI (4', 6-diamidino-2-phenylindole, Novus Biologicals), for 15 min at RT. After that, sections were washed twice with distilled water, air dried, and a coverslip placed with FluoreGuard mounting medium (Biosystems, Switzerland).

For morphometric analysis, the HE-stained sections were scanned (NanoZoomer-XR C12000; Hamamatsu, Hamamatsu City, Japan) and analysed using the software programme Visiopharm (Visiopharm 2020.08.1.8403; Visiopharm, Hoersholm, Denmark) to quantify the area of non-aerated parenchyma and aerated parenchyma in relation to the total area (= area occupied by lung parenchyma on two sections prepared from the left lung lobes) in the sections. This was used to compare the amount of air space (as an equivalent for the gas exchange surface) in the lungs between untreated and treated animals. A first app was applied that outlined the entire lung tissue as Region Of Interest (ROI, total area). For this a Decision forest method was used and the software was trained to detect the lung tissue section (total area). Once the lung section was outlined as ROI the large bronchi and vessels were manually excluded from the ROI. Subsequently, a second app with Decision forest method was trained to detect dense parenchyma (non-ventilated) and alveolar spaces (clear spaces; ventilated area) within the ROI.

Data and material availability

The coordinates and structure factors were deposited in the wwPDB with accession nos. C5 – RBD (7OAO), H3- RBD-C1(7OAP), F2–RBD (7OAY), C5-KtRBD, (7OAU), H3-KtRBD-C1, (7OAQ). EM maps and models are deposited in the EMDB and wwPDB under accession codes Spike C5 (EMD-12777, PDB ID 7OAN). Nanobody sequences are provided in the Supplementary Table 3. The pOPINO vectors for producing nanobodies C1, C5, F2 and H3

have been deposited with Addgene (www.addgene.org) with IDs 171924, 171925, 171926, 171927.

References

- 1 Dhama, K. *et al.* Coronavirus Disease 2019-COVID-19. *Clin Microbiol Rev* **33**, doi:10.1128/CMR.00028-20 (2020).
- 2 Zost, S. J. *et al.* Potently neutralizing and protective human antibodies against SARS-CoV-2. *Nature* **584**, 443-449, doi:10.1038/s41586-020-2548-6 (2020).
- 3 Liu, L. *et al.* Potent neutralizing antibodies against multiple epitopes on SARS-CoV-2 spike. *Nature* **584**, 450-456, doi:10.1038/s41586-020-2571-7 (2020).
- 4 Ju, B. *et al.* Human neutralizing antibodies elicited by SARS-CoV-2 infection. *Nature* **584**, 115-119, doi:10.1038/s41586-020-2380-z (2020).
- 5 Cao, Y. *et al.* Potent Neutralizing Antibodies against SARS-CoV-2 Identified by High-Throughput Single-Cell Sequencing of Convalescent Patients' B Cells. *Cell* **182**, 73-84 e16, doi:10.1016/j.cell.2020.05.025 (2020).
- 6 Rogers, T. F. *et al.* Isolation of potent SARS-CoV-2 neutralizing antibodies and protection from disease in a small animal model. *Science* **369**, 956-963, doi:10.1126/science.abc7520 (2020).
- 7 Chen, F., Liu, Z. & Jiang, F. Prospects of Neutralizing Nanobodies Against SARS-CoV-2. *Front Immunol* **12**, 690742, doi:10.3389/fimmu.2021.690742 (2021).
- 8 Huo, J. *et al.* Neutralizing nanobodies bind SARS-CoV-2 spike RBD and block interaction with ACE2. *Nat Struct Mol Biol* **27**, 846-854, doi:10.1038/s41594-020-0469-6 (2020).
- 9 Wrapp, D. *et al.* Structural Basis for Potent Neutralization of Betacoronaviruses by Single-Domain Camelid Antibodies. *Cell* **181**, 1436-1441, doi:10.1016/j.cell.2020.05.047 (2020).
- 10 Schoof, M. *et al.* An ultrapotent synthetic nanobody neutralizes SARS-CoV-2 by stabilizing inactive Spike. *Science* **370**, 1473-1479, doi:10.1126/science.abe3255 (2020).
- 11 Koenig, P. A. *et al.* Structure-guided multivalent nanobodies block SARS-CoV-2 infection and suppress mutational escape. *Science*, doi:10.1126/science.abe6230 (2021).
- 12 Custodio, T. F. *et al.* Selection, biophysical and structural analysis of synthetic nanobodies that effectively neutralize SARS-CoV-2. *Nat Commun* **11**, 5588, doi:10.1038/s41467-020-19204-y (2020).
- 13 Xiang, Y. *et al.* Versatile and multivalent nanobodies efficiently neutralize SARS-CoV-2. *Science* **370**, 1479-1484, doi:10.1126/science.abe4747 (2020).
- 14 Hanke, L. *et al.* An alpaca nanobody neutralizes SARS-CoV-2 by blocking receptor interaction. *Nat Commun* **11**, 4420, doi:10.1038/s41467-020-18174-5 (2020).
- 15 Xu, J. *et al.* Nanobodies from camelid mice and llamas neutralize SARS-CoV-2 variants. *Nature*, doi:10.1038/s41586-021-03676-z (2021).
- 16 Valenzuela Nieto, G. *et al.* Potent neutralization of clinical isolates of SARS-CoV-2 D614 and G614 variants by a monomeric, sub-nanomolar affinity nanobody. *Sci Rep* **11**, 3318, doi:10.1038/s41598-021-82833-w (2021).

- 17 Pymm, P. *et al.* Nanobody cocktails potently neutralize SARS-CoV-2 D614G N501Y variant and protect mice. *Proc Natl Acad Sci U S A* **118**, doi:10.1073/pnas.2101918118 (2021).
- 18 Nambulli, S. *et al.* Inhalable Nanobody (PiN-21) prevents and treats SARS-CoV-2 infections in Syrian hamsters at ultra-low doses. *Sci Adv* **7**, doi:10.1126/sciadv.abh0319 (2021).
- 19 Cardoso, F. M. *et al.* Single-domain antibodies targeting neuraminidase protect against an H5N1 influenza virus challenge. *J Virol* **88**, 8278-8296, doi:10.1128/JVI.03178-13 (2014).
- 20 Richard, G. *et al.* In vivo neutralization of alpha-cobratoxin with high-affinity llama single-domain antibodies (VHHs) and a VHH-Fc antibody. *PLoS One* **8**, e69495, doi:10.1371/journal.pone.0069495 (2013).
- 21 Rotman, M. *et al.* Fusion of hIgG1-Fc to 111In-anti-amyloid single domain antibody fragment VHH-pa2H prolongs blood residential time in APP/PS1 mice but does not increase brain uptake. *Nucl Med Biol* **42**, 695-702, doi:10.1016/j.nucmedbio.2015.03.003 (2015).
- 22 Els Conrath, K., Lauwereys, M., Wyns, L. & Muyldermans, S. Camel single-domain antibodies as modular building units in bispecific and bivalent antibody constructs. *J Biol Chem* **276**, 7346-7350, doi:10.1074/jbc.M007734200 (2001).
- 23 Detalle, L. *et al.* Generation and Characterization of ALX-0171, a Potent Novel Therapeutic Nanobody for the Treatment of Respiratory Syncytial Virus Infection. *Antimicrob Agents Chemother* **60**, 6-13, doi:10.1128/AAC.01802-15 (2016).
- 24 Jan ter Meulen Edward N. van den Brink Leo L. M. Poon, W. E. M., Cynthia S. W. Leung Freek Cox, Chung Y. Cheung, Arjen Q. Bakker, Johannes A. Bogaards, Els van Deventer, Wolfgang Preiser, Hans Wilhelm Doerr, Vincent T. Chow4 John de Kruif, Joseph S. M. Peiris, Jaap Goudsmit. Human Monoclonal Antibody Combination against SARS Coronavirus: Synergy and Coverage of Escape Mutants. *PLoS MEDICINE* **3**, e237 (2006).
- 25 Huo, J. *et al.* Neutralizing nanobodies bind SARS-CoV-2 spike RBD and block interaction with ACE2. *Nat Struct Mol Biol*, doi:10.1038/s41594-020-0469-6 (2020).
- 26 Huo, J. *et al.* Neutralization of SARS-CoV-2 by Destruction of the Prefusion Spike. *Cell Host Microbe*, doi:10.1016/j.chom.2020.06.010 (2020).
- 27 Yuan, M. *et al.* Structural basis of a shared antibody response to SARS-CoV-2. *Science* **369**, 1119-1123, doi:10.1126/science.abd2321 (2020).
- 28 Kuan-Ying A. Huang¹, T. K. T., Ting-Hua Chen¹, Chung-Guei Huang¹, Ruth Harvey, Saira Hussain, Cheng-Pin Chen¹, Adam Harding¹, Javier Gilbert- Jaramillo¹, Xu Liu¹, Michael Knight¹, Lisa Schimanski¹, Shin-Ru Shih¹, Yi- Chun Lin, Chien-Yu Cheng¹, Shu-Hsing Cheng¹, Yhu-Chering Huang, Tzou-Yien ,Lin¹ Jia-Tsong Jan, Che Ma, William James¹, Rodney S. Daniels¹, John W. McCauley¹, Pramila Rijal¹, Alain R. Townsend. Breadth and function of antibody response to acute SARS-CoV-2 infection in humans. *Plos pathogens* **17**, e1009352 (2021).
- 29 Zhou, D. *et al.* Structural basis for the neutralization of SARS-CoV-2 by an antibody from a convalescent patient. *Nat Struct Mol Biol* **27**, 950-958, doi:10.1038/s41594-020-0480-y (2020).
- 30 Daming Zhou, W. D., Piyada Supasa, Chang Liu, Alexander J. Mentzer, Helen M. Ginn, Yuguang Zhao, Helen M.E. Duyvesteyn, Aekkachai Tuekprakhon, Rungtiwa Nutalai, Beibei Wang, Guido C. Paesen, Cesar Lopez- Camacho, Jose Slon-Campos,

- Bassam Hallis, Naomi Coombes, Kevin Bewley, Sue Charlton, Thomas S. Walter, Donal Skelly, Sheila F. Lumley, Christina Dold, Robert Levin, Tao Dong, Andrew J. Pollard, Julian C. Knight, Derrick Crook, Teresa Lambe, Elizabeth Clutterbuck, Sagida Bibi, Amy Flaxman, Mustapha Bittaye, Sandra Belij- Rammerstorfer, Sarah Gilbert, William James, Miles W. Carroll, Paul Klenerman, Eleanor Barnes, Susanna J. Dunachie, Elizabeth E. Fry, Juthathip Mongkolsupaya, Jingshan Ren, David I. Stuart, Gavin R. Screaton. Evidence of escape of SARS-CoV-2 variant B.1.351 from natural and vaccine induced sera. *Cell*, doi:<https://doi.org/10.1016/j.cell.2021.02.037> (2021).
- 31 Piyada Supasa, D. Z., Wanwisa Dejnirattisai, Chang Liu, Alexander J. Mentzer, Helen M. Ginn, Yuguang Zhao, Helen M.E. Duyvesteyn, Rungtiwa Nutalai, Aekkachai Tuekprakhon, Beibei Wang, Guido C. Paesen, Jose Slon-Campos, César López-Camacho, Bassam Hallis, Naomi Coombes, Kevin Bewley, Sue Charlton, Thomas S. Walter, Eleanor Barnes, Susanna J. Dunachie, Donal Skelly, Sheila F. Lumley, Natalie Baker, Imam Shaik, Holly Humphries, Kerry Godwin, Nick Gent, Alex Sienkiewicz, Christina Dold, Robert Levin, Tao Dong, Andrew J. Pollard, Julian C. Knight, Paul Klenerman, Derrick Crook, Teresa Lambe, Elizabeth Clutterbuck, Sagida Bibi, Amy Flaxman, Mustapha Bittaye, Sandra Belij-Rammerstorfer, Sarah Gilbert, David R. Hall, Mark A. Williams, Neil G. Paterson, William James, Miles W. Carroll, Elizabeth E. Fry, Juthathip Mongkolsapaya, Jingshan Ren, David I. Stuart, Gavin R. Screaton. Reduced neutralization of SARS-CoV-2 B.1.1.7 variant by convalescent and vaccine sera. *Cell*, doi:<https://doi.org/10.1016/j.cell.2021.02.033> (2021).
- 32 Luan, B., Wang, H. & Huynh, T. Enhanced binding of the N501Y-mutated SARS-CoV-2 spike protein to the human ACE2 receptor: insights from molecular dynamics simulations. *FEBS Lett* **595**, 1454-1461, doi:10.1002/1873-3468.14076 (2021).
- 33 Greaney, A. J. *et al.* Complete Mapping of Mutations to the SARS-CoV-2 Spike Receptor-Binding Domain that Escape Antibody Recognition. *Cell Host Microbe* **29**, 44-57 e49, doi:10.1016/j.chom.2020.11.007 (2021).
- 34 Wrapp, D. *et al.* Cryo-EM structure of the 2019-nCoV spike in the prefusion conformation. *Science* **367**, 1260-1263, doi:10.1126/science.abb2507 (2020).
- 35 Walls, A. C. *et al.* Structure, Function, and Antigenicity of the SARS-CoV-2 Spike Glycoprotein. *Cell* **181**, 281-292 e286, doi:10.1016/j.cell.2020.02.058 (2020).
- 36 Wrobel, A. G. *et al.* SARS-CoV-2 and bat RaTG13 spike glycoprotein structures inform on virus evolution and furin-cleavage effects. *Nat Struct Mol Biol* **27**, 763-767, doi:10.1038/s41594-020-0468-7 (2020).
- 37 He, L. *et al.* Enhanced Ability of Oligomeric Nanobodies Targeting MERS Coronavirus Receptor-Binding Domain. *Viruses* **11**, doi:10.3390/v11020166 (2019).
- 38 Hultberg, A. *et al.* Llama-derived single domain antibodies to build multivalent, superpotent and broadened neutralizing anti-viral molecules. *PLoS One* **6**, e17665, doi:10.1371/journal.pone.0017665 (2011).
- 39 Koenig, P. A. *et al.* Structure-guided multivalent nanobodies block SARS-CoV-2 infection and suppress mutational escape. *Science* **371**, doi:10.1126/science.abe6230 (2021).
- 40 Kondo, T. *et al.* Antibody-like proteins that capture and neutralize SARS-CoV-2. *Sci Adv* **6**, doi:10.1126/sciadv.abd3916 (2020).

- 41 Chan, J. F. *et al.* Simulation of the Clinical and Pathological Manifestations of Coronavirus Disease 2019 (COVID-19) in a Golden Syrian Hamster Model: Implications for Disease Pathogenesis and Transmissibility. *Clin Infect Dis* **71**, 2428-2446, doi:10.1093/cid/ciaa325 (2020).
- 42 Imai, M. *et al.* Syrian hamsters as a small animal model for SARS-CoV-2 infection and countermeasure development. *Proc Natl Acad Sci U S A* **117**, 16587-16595, doi:10.1073/pnas.2009799117 (2020).
- 43 Sia, S. F. *et al.* Pathogenesis and transmission of SARS-CoV-2 in golden hamsters. *Nature* **583**, 834-838, doi:10.1038/s41586-020-2342-5 (2020).
- 44 Roberts, A. *et al.* Severe acute respiratory syndrome coronavirus infection of golden Syrian hamsters. *J Virol* **79**, 503-511, doi:10.1128/JVI.79.1.503-511.2005 (2005).
- 45 Zhao, G. *et al.* A Novel Nanobody Targeting Middle East Respiratory Syndrome Coronavirus (MERS-CoV) Receptor-Binding Domain Has Potent Cross-Neutralizing Activity and Protective Efficacy against MERS-CoV. *J Virol* **92**, doi:10.1128/JVI.00837-18 (2018).
- 46 Liang, W., Pan, H. W., Vllasaliu, D. & Lam, J. K. W. Pulmonary Delivery of Biological Drugs. *Pharmaceutics* **12**, doi:10.3390/pharmaceutics12111025 (2020).
- 47 Yuan, M. *et al.* A highly conserved cryptic epitope in the receptor binding domains of SARS-CoV-2 and SARS-CoV. *Science* **368**, 630-633, doi:10.1126/science.abb7269 (2020).
- 48 Wrapp, D. *et al.* Structural Basis for Potent Neutralization of Betacoronaviruses by Single-Domain Camelid Antibodies. *Cell* **181**, 1004-1015 e1015, doi:10.1016/j.cell.2020.04.031 (2020).
- 49 Van Heeke, G. *et al.* Nanobodies(R) as inhaled biotherapeutics for lung diseases. *Pharmacol Ther* **169**, 47-56, doi:10.1016/j.pharmthera.2016.06.012 (2017).
- 50 Esparza, T. J., Martin, N. P., Anderson, G. P., Goldman, E. R. & Brody, D. L. High affinity nanobodies block SARS-CoV-2 spike receptor binding domain interaction with human angiotensin converting enzyme. *Sci Rep* **10**, 22370, doi:10.1038/s41598-020-79036-0 (2020).
- 51 Nettleship, J. E., Rahman-Huq, N. & Owens, R. J. The production of glycoproteins by transient expression in Mammalian cells. *Methods Mol Biol* **498**, 245-263, doi:10.1007/978-1-59745-196-3_16 (2009).
- 52 Bird, L. E. *et al.* Application of In-Fusion cloning for the parallel construction of E. coli expression vectors. *Methods Mol Biol* **1116**, 209-234, doi:10.1007/978-1-62703-764-8_15 (2014).
- 53 Emsley, P. & Cowtan, K. Coot: model-building tools for molecular graphics. *Acta Crystallogr D Biol Crystallogr* **60**, 2126-2132, doi:10.1107/S0907444904019158 (2004).
- 54 Liebschner, D. *et al.* Macromolecular structure determination using X-rays, neutrons and electrons: recent developments in Phenix. *Acta Crystallogr D Struct Biol* **75**, 861-877, doi:10.1107/S2059798319011471 (2019).
- 55 Williams, C. J. *et al.* MolProbity: More and better reference data for improved all-atom structure validation. *Protein Sci* **27**, 293-315, doi:10.1002/pro.3330 (2018).
- 56 Afonine, P. V. *et al.* New tools for the analysis and validation of cryo-EM maps and atomic models. *Acta Crystallogr D Struct Biol* **74**, 814-840, doi:10.1107/S2059798318009324 (2018).

- 57 Bewley, K. R. *et al.* Quantification of SARS-CoV-2 neutralizing antibody by wild-type plaque reduction neutralization, microneutralization and pseudotyped virus neutralization assays. *Nat Protoc*, doi:10.1038/s41596-021-00536-y (2021).
- 58 Wolfel, R. *et al.* Virological assessment of hospitalized patients with COVID-2019. *Nature* **581**, 465-469, doi:10.1038/s41586-020-2196-x (2020).
- 59 Patterson, E. I. *et al.* Methods of Inactivation of SARS-CoV-2 for Downstream Biological Assays. *J. Infect. Dis.* **222**, 1462-1467, doi:10.1093/infdis/jiaa507 (2020).

Acknowledgements

This work was supported by the Rosalind Franklin Institute, funding delivery partner EPSRC. PPUK is funded by the Rosalind Franklin Institute EPSRC grant no. EP/ S025243/1. J.H.N., A.L.B., P.J.H., M.W. and P.W. are supported by Wellcome Trust (100209/Z/12/Z). J.H. is supported by the EPA Cephalosporin Fund. X-ray data were obtained using Diamond Light Source COVID-19 Rapid Access time on Beamline I03, I04 and I24 (proposal MX27031). The core virus neutralization facility is supported by gifts to the Oxford COVID-19 Research Response Fund. EM results were obtained at the national EM facility at Diamond, eBIC, through rapid access proposal BI27051. Work at the University of Liverpool is supported by MRC grant MR/W005611/1, G2P-UK; A National Virology Consortium to address phenotypic consequences of SARSCoV-2 genomic variation (JPS and JAH) and by the US Food and Drug Administration (USA) 75F40120C00085, Characterization of severe coronavirus infection in humans and model systems for medical countermeasure development and evaluation (JAH). We wish to thank the laboratory staff of the Histology Laboratory, Institute of Veterinary Pathology, Vetsuisse Faculty, University of Zurich, and the laboratory staff of the Pathology Laboratory and Biological Investigations Group Public Health England, Porton Down for excellent technical support. We are grateful to Josep Monné Rodriguez for his assistance in the design of the apps for the morphometric assessment. We thank Tomas Malinauskas (Oxford University) and colleagues at the CMB (Oxford

University) for assistance with protein production and Professor Gary Stephens, Barney Jones and Hong Lin (Reading University) for expertise in llama immunisation.

Contributions

J.H. isolated the nanobodies, designed trimers and carried out SPR analyses. M.W. and D.K.C. performed the EM studies. H.M., L.M., A.L.B, J.H. and J.H.N. performed the crystallography and ITC experiments. J.H., A.L.B., J.D., C.N., P.J.H., P.N.W., M.D. produced proteins for the experiments. A.H., L.B, K.R.B, M.J.E and W.J. carried out neutralization assays and analysis. J.J.C., P.S., Y.H. and S.F. carried out the animal study. R.W., O.C., D.K., D.N. carried out the molecular biology and live viral assays. A.K. and F.J.S. performed pathological analyses, immunohistology and morphometric analyses. J.P.S., J.T. M.C. directed the animal studies R.J.O. and J.H.N. planned the project and wrote the manuscript with contributions from all authors.

Competing interests

The Rosalind Franklin Institute has filed a patent that includes the four nanobodies described here, R.J.O, J.H. and J.H.N., are named as inventors. The other authors declare no competing interests.

Figures legends

Fig. 1. Nanobody binding kinetics

(a-d) SPR sensorgrams showing binding kinetics of nanobody C5, H3, C1 and F2 for RBD Victoria (immobilized as biotinylated RBD on the chip), **(e-g)** SPR sensorgrams of competition assays between RBD and C5, H3, C1, F2 for binding to **(e)** ACE-2 **(f)** CR3022 and **(g)** H11-H4, with all ligands immobilised as Fc fusion proteins and C2Nb6 (an anti-Caspr2 nanobody)

used as a negative control, **(h -j)** binding kinetics of nanobody C5, H3, C1 and F2 for RBD Kent (immobilized as biotinylated RBD on the chip).

Fig. 2. Crystal structures of nanobody-RBD complexes

(a) The four nanobodies of this study are shown in cartoon and labelled. The figure was generated by superimposing the RBD protein from each crystal structure, only one RBD monomer is shown. Also shown is ACE2 (cyan surface) from the RBD ACE2 complex (PDB 6M0J), positioned by superposition of the RBD. Nanobodies C5 and H3 compete with ACE2 for binding to RBD. F2 and C1 bind to a different epitope, although a loop of C1 (G42) would clash with ACE2 (arrow). **(b)** RBD is shown as a surface, the RBD molecule has been rotated by 90 ° relative to (a). The surface is colored magenta corresponds to the epitope engaged by both C1 and F2, in red is the additional region recognized by C1 only. In yellow is the epitope recognized by C3 only, in black by H3 only and in green by both C5 and H3. **(c)** The same molecule and color scheme as (b) but rotated by 90 ° to more clearly show the H3 and C5 epitopes. The key molecular interactions between **(d)** C5, **(e)** H3 **(f)** C1 and **(g)** F2 and RBD are identified and labelled. RBD is in approximately in the same orientation as **(a)**. In (f) and (g) coloured in magenta and gold respectively is the portion of RBD that is also recognised by both C1 and F2. **(h)** C1 and F2 bind to RBD in different orientations and overlap at residues 102 and 103. Their spatial relationship can be described as an approximate 40 ° rotation around the main chain at 102 and 103. **(i)** In the F2 (blue) RBD (cyan) complex, Y102 of F2 results in a displacement of the helix at Y369 of RBD relative to the C1 (red) and RBD (brown) complex. The orientation of the molecules are the same as shown in Figure 2a.

All structural figures were prepared using PyMOL (<http://www.pymol.org/>).

Fig. 3 Comparison of nanobody-RBD complexes

(a) Superimposition of H11-H4-RBD and H3-RBD complexes; V102 is shown by a red sphere. (b) Overlay showing the key salt bridge interaction between E484 in RBD and R31 in nanobody H3 and R52 in nanobody C5, respectively. (c) Close-up of the RBD-C5 interfaces for complexes with the Victoria strain of SARS-CoV-2 (N501; left hand side) and Alpha strain (N501Y; right hand side) showing the hydrogen bonding between N501 and Y501 of RBD (coloured green) with N73 of C5 in yellow and wheat respectively. Key residues are shown in stick representations.

Fig. 4. Cryo-EM structure of C5-Spike complex

(a) EM structure of spike (S1) trimer with each of three chains bound to one C5 nanobody coloured yellow. The other spike monomers are colored pale cyan, green and purple wheat and throughout and show that all three RBDs are in the ‘down’ conformation. (b) Superimposition of C5 onto the spike protein in the ‘two down one up’ conformation shows that there would be significant clashes that would prevent this interaction.

Fig. 5. Neutralisation of SARS-CoV-2 strains in vitro

Neutralisation curves of the anti-RBD nanobody trimers for (a) Victoria (B) (b) Kent (B1.1.7) and (c) South Africa (B1.351) strains of SARS-CoV-2 measured in a microneutralisation assay. Data are shown as the mean +/- 95% CI.

Fig. 6. C5-Fc Neutralisation of SARS-CoV-2 in the Syrian hamster model.

(a) Animals were challenged with SARS-CoV-2 (B Victoria 5×10^4 pfu) at day 0 and then treated with either C5-Fc (IP 4 mg/kg) or PBS, delivered by the intraperitoneal route 24 hours post-challenge and Throat Swab (TS) and Nasal Wash (NS) samples collected on days 2, 4, 6

and 7 post virus challenge. **(b)** Body weight was recorded daily and the mean percentage weight change from baseline was plotted (± 1 SE). Filled in square represents data from control animals (virus only) and filled in circles represents data from nanobody treated. Nasal washes (i-iii) and oropharyngeal swabs (iv-vi) were collected at days -2 to 2, 4, 6 and 7 pc for all virus challenged groups. Tissue samples (lung, trachea and duodenum) were collected at post-mortem (day 7 pc) (vii & viii). Open square represents data from control animals (virus only) and open circle represents data from nanobody treated hamsters. Symbols show values for individual animals, columns represent the calculated group geometric means. **(c)** quantitation of live virus in the nasal wash and oropharyngeal swabs using a micro-foci assay **(d)** number of copies of subgenomic (sg)viral RNA in the nasal wash and oropharyngeal swab **(e)** number of copies genomic viral RNA in the nasal wash oropharyngeal swab. **(f)** number of copies of sgRNA and genomic RNA in tissues. The dashed horizontal lines show the lower limit of quantification (LLOQ) and the lower limit of detection (LLOD).

Mann-Whitney's U test for median comparisons.

Figure 7. Therapeutic efficacy of C5 Trimer in Syrian hamster model.

(a) Golden Syrian hamsters ($n = 6$ per group) were infected intranasally with SARS-CoV-2 strain LIV (PANGO lineage B; 10^4 pfu). Individual cohorts were treated either 2h pre-infection or 24 h post-infection (hpi) with 100 μ l of C5 either intranasally (IN) or intraperitoneally (IP) as indicated or sham-infected with PBS. **(b)** Animals were monitored for weight loss at indicated time-points. Data are the mean value \pm SEM. Comparisons were made using a repeated-measures two-way ANOVA. ** represents $p < 0.01$. **(c)** RNA extracted from lungs was analysed for SARS-CoV-2 viral load using qRT-PCR for the N gene levels by qRT-PCR. Assays were normalised relative to levels of 18S RNA. Data for individual animals are shown with the median value represented by a horizontal line. Comparisons were made using a Mann-

Whitney U test ** represents $p < 0.01$ and * represents $p < 0.05$. **(d)** Morphometric analysis of HE-stained sections scanned and analysed using the software programme Visiopharm to quantify the area of non-aerated parenchyma and aerated parenchyma in relation to the total area. Results are expressed as the mean free airspace in lung sections. Pairwise comparisons were made between groups using a Mann-Whitney U test * represents $p < 0.05$; ** represents $p < 0.01$. **(e)** Lung sections of hamsters, infected intranasally with 10^4 PFU/100 μ l SARS-CoV-2 and euthanized at day 7 post infection. Animals had been untreated prior to infection (PBS) or treated with 4 mg/kg C5 IN 2 h prae infection (h prae inf) or 24 h post infection (h post inf) or IP at 24 h post inf, or had received 0.4 mg/kg C5 IN at 24 h post inf. In the untreated animal (PBS) the lung parenchyma exhibits a large consolidated area (arrow) and multifocal patches with extensive viral antigen expression in particular by pneumocytes. In treated animals there are only a few small areas of consolidation (arrows). The animal treated with 4 mg/kg C5 intranasally at 2 h prae inf exhibits a few small patches with viral antigen expression mainly in degenerate cells, all other treated animals show viral antigen expression in occasional individual macrophages within small infiltrates or in pneumocytes in individual alveoli. Top: HE stain, bottom: immunohistology for SARS-CoV-2 N, hematoxylin counterstain. Bars = 20 μ m (PBS) or 10 μ m (all others).

Figure RR1 TEM screening images of Spike in complex with C5 trimer (molar ratio 1:1.5). (a) The grid that was screened. (b) Two grid squares were analysed in detail. Illustrative images taken at grid square magnification, hole magnification and data acquisition magnification are shown. We did not identify promising single particles for further study, rather it appeared to be ill defined aggregates.

Figure RR2 TEM screening images of Spike in complex with C5 trimer (molar ratio 1:3). (a) The grid that was screened. (b) Two grid squares were analysed in detail. Illustrative images taken at grid square magnification, hole magnification and data acquisition magnification are shown. We did not identify promising single particles for further study, rather it appeared to be ill defined aggregates.

REVIEWER COMMENTS

Reviewer #1 (Remarks to the Author):

This revision well addressed my concerns. I recommend publishing without further revision.